# Single cell profiling of primary and paired metastatic lymph node tumors in breast cancer patients

Tong Liu[1,7], Cheng Liu[2,3,4,7], Meisi Yan[5,7], Lei Zhang[2,3,4], Jing Zhang[2,3,4], Min Xiao[1], Zhigao Li [1] ✉, Xiaofan Wei [2,3,4] ✉ & Hongquan Zhang [2,3,4,6] ✉

The microenvironment of lymph node metastasized tumors (LNMT) determines tumor progression and response to therapy, but a systematic study of LNMT is lacking. Here, we generate single-cell maps of primary tumors (PTs) and paired LNMTs in 8 breast cancer patients. We demonstrate that the activation, cytotoxicity, and proliferation of T cells are suppressed in LNMT compared with PT. $CD4^+CXCL13^+$ T cells in LNMT are more likely to differentiate into an exhausted state. Interestingly, $LAMP3^+$ dendritic cells in LNMT display lower T cell priming and activating ability than in PT. Additionally, we identify a subtype of $PLA2G2A^+$ cancer-associated fibroblasts enriched in $HER2^+$ breast cancer patients that promotes immune infiltration. We also show that the antigen-presentation pathway is downregulated in malignant cells of the metastatic lymph node. Altogether, we characterize the microenvironment of LNMT and PT, which may shed light on the individualized therapeutic strategies for breast cancer patients with lymph node metastasis.

Metastasis is the most prominent cause of cancer morbidity and mortality. Due to the special structure of lymph node (LN) vessels in tumors, tumor cells tend to metastasize to LN tissue, and tumor cell metastasis to LNs is an early manifestation of metastatic tumors[1]. The microenvironment of lymph node metastasized tumors (LNMTs) is considered to be immunosuppressive; however, the characteristics and specific mechanisms of various immune cells are not clear. There is an apparent need for the therapeutic targeting of LNMT, rather than of the PT, to secure proper antitumor T-cell generation and timely tumor infiltration[2,3]. Studies have found that metastatic LNs can affect the immune response of tumors. For example, targeting metastatic LNs can significantly enhance the therapeutic effect on PTs[4], which may represent an important strategy for improving patient survival.

Therefore, it is important to characterize and differentiate between the LNMT and PT microenvironments.

Genomic and transcriptomics technologies can help us to understand the changes of tumor cells during metastasis to distant organs[5]. In recent years, the emergence of and advances in single-cell sequencing technology have provided a new and precise approach for understanding the complexity of genetic heterogeneity in tumor evolution and tumor cell metastatic progression[6]. Tumor cells in a state of partial epithelial–mesenchymal transition located in the periphery of the original tumor are more likely to metastasize than are the cells inside the tumor[7]. It is well known that tumors form as a consequence of interactions between malignant cells and the microenvironment[8].

[1]Department of Breast Surgery, Harbin Medical University Cancer Hospital, Harbin, China; Heilongjiang Academy of Medical Sciences, Harbin, China. [2]Program for Cancer and Cell Biology, Department of Human Anatomy, Histology and Embryology, School of Basic Medical Sciences, Peking University Health Science Center, Beijing 100191, China. [3]Peking University International Cancer Institute, Peking University Health Science Center, Beijing 100191, China. [4]MOE Key Laboratory of Carcinogenesis and Translational Research and State Key Laboratory of Natural and Biomimetic Drugs, Peking University Health Science Center, Beijing 100191, China. [5]Department of Pathology, Harbin Medical University, Harbin 150081, China. [6]Department of Human Anatomy, Histology, and Embryology, Shenzhen University School of Medicine, Shenzhen 518055, China. [7]These authors contributed equally: Tong Liu, Cheng Liu, Meisi Yan. ✉e-mail: drzhigaoli@hrbmu.edu.cn; weixiaofan@bjmu.edu.cn; Hongquan.Zhang@bjmu.edu.cn

Breast cancer is the most common cancer in women worldwide and has a high mortality rate[9]. The molecular characteristics of breast cancer vary considerably among the different subtypes, and specific therapeutic approaches are required for each classification[10]. Therefore, understanding the composition of different types of breast cancer has great clinical value. Thus far, several studies have used single-cell sequencing to investigate breast cancer. For instance, one study explored the mechanism of treatment resistance of malignant cells in triple-negative breast cancer (TNBC) by using single-cell copy number variation (CNV) sequencing technology[11], while another study on TNBC characterized tumor-infiltrating immune cells[12]. Research involving single-cell analysis of breast cancer has mainly centered on immune cells or tumor cells, but there is currently a lack of systematic research on the interactions among various cell types, and the relevant published articles have primarily focused on TNBC[11,13,14]. It has further been shown that luminal and human epidermal growth factor receptor 2 (HER2)[+] patients have a higher risk of LN metastasis than patients with TNBC[15]. Therefore, we chose to direct the focus of our study on non–basal-like breast cancer.

In this work, we aim to investigate the regulatory mechanism of the tumor microenvironment (TME) that may contribute to malignant cell metastasis and the colonization of LNs, with a particular focus on uncovering the differences between the PT and LNMT in non-TNBC patients using single-cell RNA sequencing. Our analyses reveal that the microenvironment of metastatic LNs is more conducive to tumor cell survival than is that of tumors in situ due to the lower immune cell activity of metastatic LNs. Moreover, we identify a type of cancer-associated fibroblast (CAF) expressing *PLA2G2A* that can interact with immune cells and that is enriched in HER2[+] breast cancer. Our data provide insights into the mechanism of LNMT and immune infiltration in tumors and may be a valuable reference for the clinical application of immunotherapy for breast cancer metastasis.

## Results

### The microenvironmental landscape of PT and LNMT in breast cancer

To characterize the TME of the PT and LNMT in patients with breast cancer, we collected paired tissues of LNMT and PT from 8 treatment-naïve patients with breast cancer subtypes including luminal A, luminal B, and HER2[+]. These tissues were separated into single cells, and we obtained a total of 118,845 cells sequenced by using 10x Genomics 5′ mRNA and T cell receptor (TCR) sequencing methods (Fig. 1a). Hematoxylin and eosin (HE) staining showed the gross appearance of metastatic LNs, indicating a high frequency of LNMT among the enrolled patients (Supplementary Fig. 1a). We used BBKNN integration to integrate cells from different patients[16] (Supplementary Fig. 1b). All of the cells could be divided into the following 9 major types according to their canonical markers: B cells (*CD3D*, *CD79A*), CD4 T cells (*CD3D*, *CD4*), CD8 T cells (*CD3D*, *CD8A*), NK cells (*GNLY*), myeloid cells (*LYZ*), epithelial cells (*EPCAM* and *KRT19*), CAFs (*PDGFRA*), perivascular-like (PVL) cells (*RGS5*), and TECs; (*PLVAP*; Supplementary Fig. 1c−e). We found that the cell types in PT varied largely across the patients, but those in LNMT were similar (Supplementary Fig. 1f). Then, we further clustered and annotated the cells into 40 different cell clusters according to their specific marker genes (Fig. 1b, Supplementary data 1). To investigate the enrichment of cell types in PT and LNMT, we calculated the percentage of the tissues within each cluster. The data showed that B cells and CD4 T cells, which are well-known cellular components of normal LNs, are enriched in the microenvironment of LNMT. In contrast, epithelial cells, CD8 T cells, CAFs, and mast cells were abundant in PT (Fig. 1c, d). The difference in cell types between PT and LNMT was also shown in the UMAP embedding plot which was colored according to tissue type (Fig. 1e). To further prove that the cells we collected were intratumoral, we randomly picked LN tissues of 4 patients from the 8 patients in the present study

to perform spatial transcriptomics. The results showed that the ratio of tumor cells was over 50% in all 4 patients and over 75% in 3 of them, suggesting that the samples were from LN metastases (Fig. 1f, Supplementary Fig. 1g). These data indicated that the cell types of the metastatic microenvironment of LNs differed significantly from those of PTs.

### Suppressed T cell activity in LNMTs

LNs are central to immune cell circulation and maturation. To determine why malignant cells can survive in LNs without being eliminated by the immune cells, we analyzed the features of immune cells both in PTs and LNMTs. We annotated T cells and NK cells into 15 clusters, including 5 CD8 T cell subsets, 7 CD4 T cell subsets, γδ T cells, and 2 NK cell subsets (Fig. 2a, Supplementary Fig. a & b). The data showed that T cells and NK cells from the 2 tissues were distributed differently and exhibited disparate transcription programs (Fig. 2a). We found that these 2 types of NK cells expressed tumor-suppressing genes (*XCL1* and *XCL2*[17] for NK-C1-*XCL1*, cytotoxic genes for NK-C2-*GZMH*) in higher proportions in PTs than in LNMTs (Supplementary Fig. 2c).

To track the development trajectories of CD8 T cells, we employed a diffusion embedding map to visualize CD8 T subsets and found continuous developmental progression (Fig. 2b). CD8-C1-*CD8B* was present at the initial stage of CD8 T cells differentiation, in which there was also a high expression of *CCR7* and *SELL* (Fig. 2b, Supplementary Fig. 2c), which are the markers of naïve T cells ($T_N$). CD8-C2-*CCL5*, characterized by higher expression of cytotoxic markers and high expression of *HOPX* (Supplementary Fig. 2b, d), was present in the next stage after CD8-C1-*CD8B*. *CCL5*[+] T cells further branched into CD8-C3-*GZMK* or CD8-C4-*HSPA1A* (Fig. 2b). CD8-C3-*GZMK* cells were defined as effect memory T cells ($T_{EM}$) due to their expression of cytotoxic markers like *NKG7*, *GZMA*, and *GZMK*; meanwhile, CD8-C4-*HSP1A1*, which expressed cytotoxic markers and a high level of *CD69* but a low level of *ITGAE*, represented *CD69*[+]*ITGAE*[-] tissue-resident memory T cells ($T_{RM}$) (Fig. 2b, Supplementary Fig. 2b). CD8-C5-*CXCL13*, which differentiated from CD8-C3-*GZMK*, was considered to be the terminal state of differentiation (Fig. 2b). CD8-C5-*CXCL13* expressed cytotoxic markers and exhausted makers including *CTLA4*, *PDCD1*, and *LAG3* (Supplementary Fig. 2b, e), and was characterized as exhausted or pre-exhausted CD8 T cells. We then performed principal component analysis (PCA) to investigate CD8 T cells in the microenvironment of LNMTs and PTs. Principal component (PC) 2 was the most prominent component distinguishing CD8 T cells between PT and LNMT of the first 20 PCs (Supplementary Fig. 2f). We used a PCA embedding map colored according to tissue type to determine the distribution of CD8 T cells (Fig. 2c). The most variant genes contributing to PC2 were *CCL5*, *CCL4*, chemokines involved in T cell recruitment, MHC class II genes functioning in antigen presentation, and cytokines such as *GZMA* and *GZMH*, which are known for their cytotoxic function in T cells (Supplementary Fig. 2g). The CD8 T cell activation signature also showed CD8 T cells in PTs to have a higher activation score than those in LNMTs (Fig. 2d), supporting the previous findings.

Next, we analyzed the characteristics of CD4 T cells. Single-cell sequencing has recently been used to identify the tumor-suppressing functions of CD4 T cells, for example, the function of *GZMK*[+]*CD4*[+] T cells in bladder cancer[18]. It has been found that CD4 T cells are more complex than initially believed and need to be further explored. In our study, CD4 T cells were classified into 7 clusters according to their specific gene expressions (Fig. 2a, Supplementary Fig. 2b). According to their markers, CD4-C1-*RPL* corresponded to naïve CD4 cells, while CD4-C2-*ANXA1* and CD4-C3-*YPEL5* corresponded to memory or pre-memory CD4 cells. These 3 clusters were enriched in LNMTs (Fig. 2a, Supplementary Fig. 2c). CD4-C6-*CXCL13* highly expressed *CXCL13*, which is reported to attract B cells[19], and corresponded to $T_{fh}$ or $T_{fh}$-like cells. Interestingly, we found notable heterogeneity among *CD4*[+]*CXCL13*[+] T cells across the 2 microenvironments (Fig. 2e, f). To

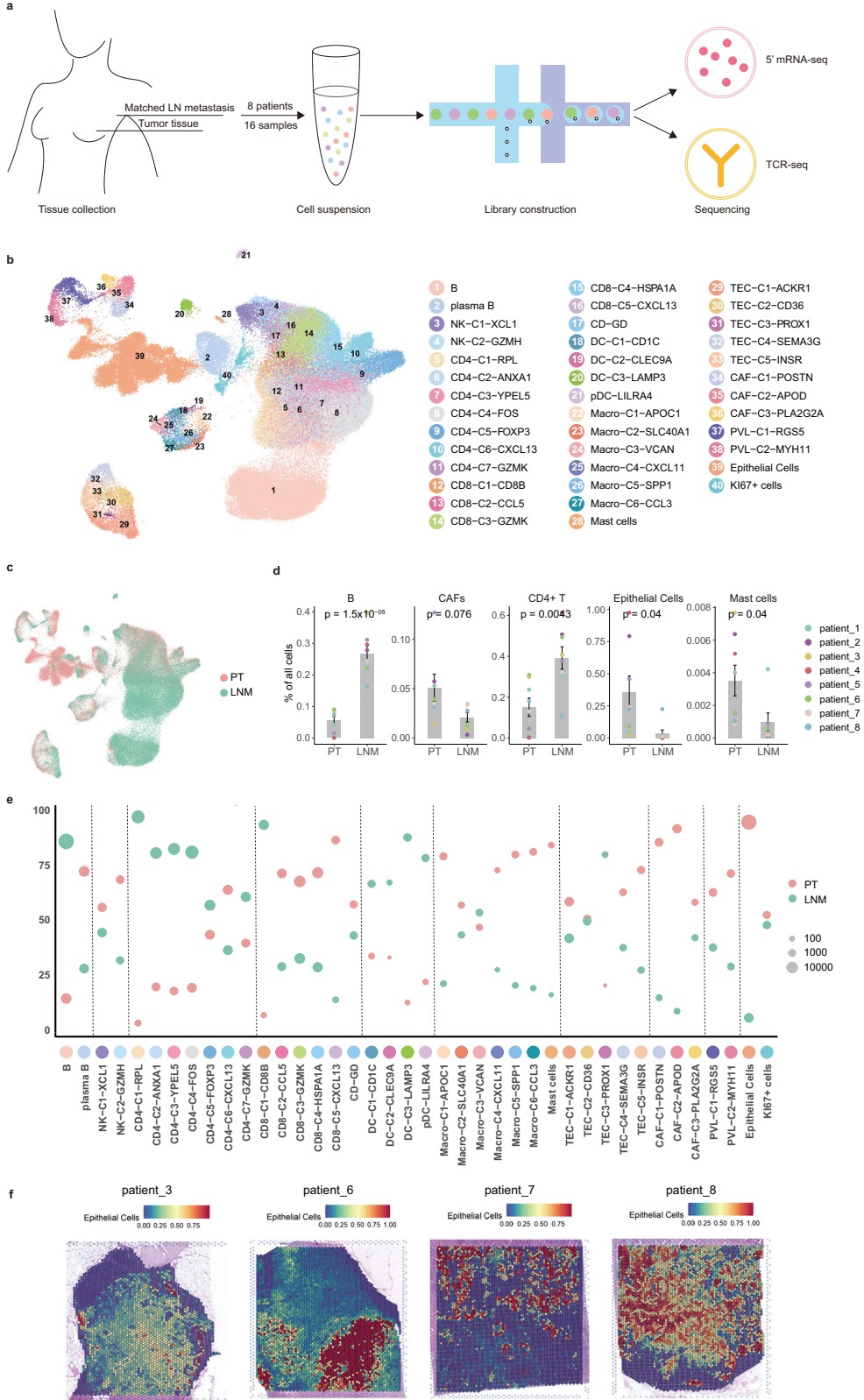

**Fig. 1 | Microenvironmental landscape of PT and LNMT in breast cancer.**
**a** Diagram of the single-cell sequencing strategy for lymph node metastasis patients. **b, c** UMAP embedding plot showing identified clusters of all 118845 cells from paired PT (primary tumor) and LNMT (lymph node metastasized tumors) of 8 LNMT patients. Cells were colored according to their clusters (**b**) or tissues (**c**). The number of cells per cluster, per patient, and per tissue is summarized in Supplementary data 2. **d** Bar plots showing the differences in the major cell types between

the 2 tissues (PT: $n = 8$ samples, LNM: $n = 8$ samples). Statistical testing was performed by a two-sided Wilcoxon test. Data are presented as mean values+/− SD. **e** The relative proportion of clusters between breast tumor and metastasis lymph node samples. Cell proportion has been normalized by sample size. **f** Spatial transcriptome analysis revealed the distribution of epithelial cells in the LNMT of 4 patients. Source data are provided as a Source Data file.

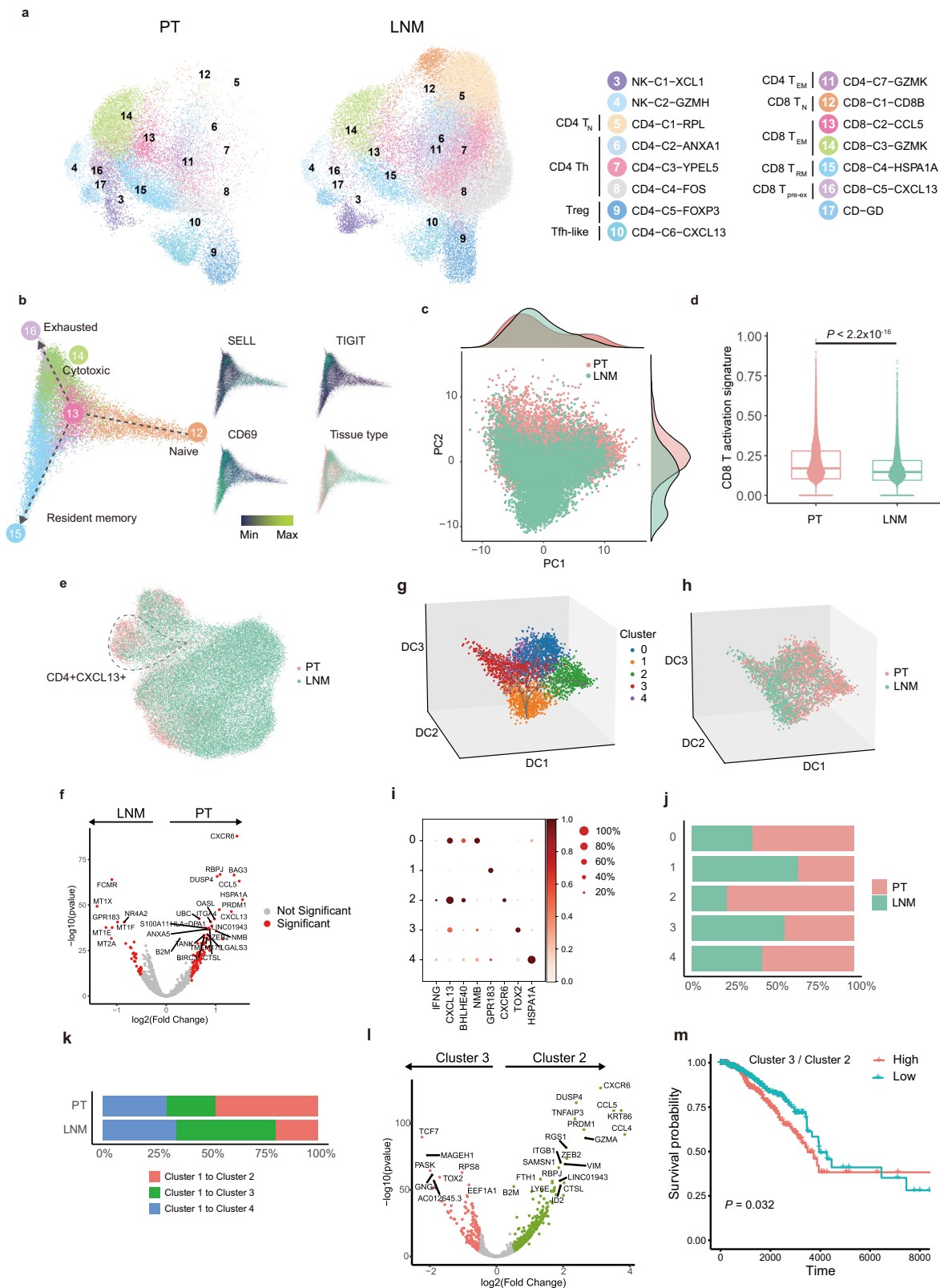

study the heterogeneity of *CD4⁺CXCL13⁺* cells, we re-clustered *CD4⁺CXCL13⁺* T cells and found 5 clusters of *CD4⁺CXCL13⁺* T cells as identified by their typical markers (Fig. 2g–i). We found that the percentage of cluster 2 in PTs was much higher than that in LNMTs (Fig. 2j). Cluster 2 expressed a higher level of interferon-gamma (IFN-γ) than did the other *CD4⁺CXCL13⁺* clusters, indicating that this type of CD4 T cells has a potential role in killing tumor cells (Fig. 2i).

Intriguingly, we found that *BHLHE40* was highly expressed in cluster 2 (Fig. 2i), which is consistent with a previous study that reported *BHLHE40⁺CD4⁺* T cells to have the ability to suppress colon cancer cells[20], supporting the speculation of the tumor-suppressing function of cluster 2, which need further functional validation. In contrast, LNMTs showed more cluster 1 cells with a high expression of *GPR183*, which is reportedly expressed in naive CD4 and CD8 T cells[21] (Fig. 2i).

**Fig. 2 | T cell activity suppression in LNMT. a** A UMAP embedding plot of 58,316 T and NK cells grouped into 15 clusters from PTs and LNMTs of 8 patients. **b** The developmental trajectory of CD8 T cells is inferred by the diffusion map and colored according to tissue type and expression of example genes. **c** Scatter plot showing PCA components of CD8 T cells color-coded according to tissue type. The top density plot displays the distribution of CD8 T cells along PC1, the right density plot displays the distribution of CD8 T cells along PC2. **d** A box plot showing a comparison of the CD8 T activation score across tissues (PT: n = 9224 cells, LNMT: n = 7275 cells) as calculated by the normalized gene mean expression of T activation signature genes (Supplementary data 3) in CD8 T cells. Statistical testing was performed by a two-sided Wilcoxon test. In the box plots, the center line corresponds to the median, box corresponds to the interquartile range (IQR), and whiskers 1.5 × IQR. **e** A UMAP embedding plot showing CD4 clusters color-coded according to tissue type; the CD4-C6-*CXCL13* cluster is circled. **f** A volcano plot showing the differentially expressed genes between PTs and LNMTs in the CD4-C6-*CXCL13* subset. P value <0.05, log2 (fold change) ≥ 0.5. Statistical testing was performed by a two-sided Wilcoxon test. The *P*-values were corrected with Benjamini-Hochberg

adjustment. **g, h** 3D projection of the CD4-C6-*CXCL13* cluster inferred by a diffusion map and colored according to the subset (**f**) and tissue type (**g**). **i** A dot plot showing marker genes across 5 subsets of CD4-C6-*CXCL13*. The dot size indicates the fraction of expressing cells, and dots are colored according to normalized z score expression. **j** The proportion of tissues within 5 subsets in CD4-C6-*CXCL13*; the coloring is according to tissue type; the *x*-axis represents the fraction of tissues, and the *y*-axis represents major cell types. **k** Percentages of the differentiation direction of cluster 1 to other clusters in 2 microenvironments. **l** Volcano plot showing differentially expressed genes between cluster 2 and cluster 3. P value < 0.05, log2 (fold change) ≥ 0.5. Statistical testing was performed by a two-sided Wilcoxon test. The *P* values were corrected with Benjamini-Hochberg adjustment. **m** Kaplan-Meier plot shows that patients with breast cancer in the TCGA dataset (HER2+: n = 67 samples, LumA: n = 421 samples, LumB: n = 192 samples) with a high ratio of cluster 3 to cluster 2 have shorter overall survival. The ratio of cluster 3 to cluster 2 is the average expression of the cluster 3 signature genes divided by the average expression level of the cluster 2 signature genes. Statistical testing was performed by Log-Rank test. Source data are provided as a Source Data file.

Using a diffusion map to infer the *CD4⁺CXCL13⁺* T cell trajectory, we found that the initial development stage of cluster 1 could differentiate into 3 branches: cluster 2, cluster 3, and cluster 4 (Fig. 2g). We found that cluster 1 in LNMT mainly differentiated into exhausted cluster 3, whereas in PT, it mainly differentiated into tumor-suppressing cluster 2 (Fig. 2k). Differential gene analysis between cluster 2 and cluster 3 confirmed that cluster 2 expressed high tumor-suppressing genes (e.g., *GZMA*) in tumor (Fig. 2l). The ratio of cluster 3 to cluster 2 could also be used to predict poor prognosis in TCGA-BRCA datasets (Fig. 2m). This finding suggested that CD4 T cells in LNMTs are less mature than those in PTs and *CD4⁺CXCL13⁺* T cells and are more likely to be reprogrammed into an exhausted state in LNMTs. Altogether, these data demonstrated that the antitumor cytotoxicity of T cells in LNMTs was reduced compared to that in PTs.

## T cells in PTs showed higher expansion and transition ability than did those in LNMTs

T clonal expansion and transition are manifestations of immune response and immune activation. Consequently, we analyzed T clonal expansion and transition in CD4 and CD8 T cells with single-cell resolution TCR sequencing. 59,327 immune T cells were captured and 47,803 of them were performed TCR analysis. We found that patient 2 and patient 4 had very rare immune cells, which were removed from the downstream analysis (Supplementary Fig. 3a). Of all the T cells, approximately 18.4% showed clonal expansion (5.01% with 2 cells per clone, and 13.39% with 3 cells per clone; Supplementary Fig. 3b). We found that the proportion of T cells in the PTs with clonal expansion was higher than that in LNMTs (Fig. 3a). We then employed STARTRAC, a method based on Shannon entropy, to quantify the expansion and transition ability of T cell clusters[20]. The transition and expansion ability of CD8 T cells was more powerful than that of CD4 T cells (Supplementary Fig. 3c, d). We found that CD4 T cells in PTs, such as CD4-C3-*YPEL5*, had stronger transition ability than did those in LNMTs (Fig. 3b). Then, we used STARTRAC-expansion to measure the expansion of each T cell cluster. CD4-C6-*CXCL13* had the greatest expansion ability among CD4 T cells, followed by CD4-C5-*FOXP3* (Supplementary Fig. 3c). these both clusters showed a greater expansion ability in PTs than in LNMTs (Fig. 3c). CD8-C2-*CCL5* and CD8-C3-*GZMK* demonstrated a significantly more powerful expansion ability in PTs than in LNMTs (Fig. 3c). To investigate the relationship between T cell expansion and T cell activation based on TCR sequencing analysis, we fitted the line of CD4 regulatory T cells (Tregs) score and CD8 activation score with cell expansion in each cluster in different tissues separately to determine the correlation between T cell activation and TCR expansion. T cells showed a weaker expansion ability in LNMTs than in PTs, even when at a similar developmental stage (Fig. 3d, e). Overall, TCR sequencing analysis of CD4 and CD8 T cells further

proved that T cells displayed a lower activity of expansion and transition in LNMTs than in PTs.

## Characteristics of matched T cells in the breast cancer microenvironment

To further demonstrate that T cells are suppressed in LNMTs, we compared the activity of matched T (MT) cells in LNMTs with those in PTs. T cells with identical TCRs located in 2 different tissues were considered to be MT cells, originating from the same progenitor and having a similar development time[22] (Fig. 4a). The results showed that the microenvironment of PTs had a higher percentage of MT cells (25%) than did the LNMTs (5.8%) (Fig. 4b). Matched T cells only occupy lower than 25% in total T cells (Fig. 4b) and may transit between diverse states. To avoid missing the differences caused by different states of matched T, we included all matched CD8 T in DEG analysis and separated matched CD4 T cells into conventional CD4 T (Tconvs) and Tregs for differential gene analysis. The results showed that 128 genes were significantly down-regulated and 63 genes were up-regulated in matched CD8 T cells of LNMT compared with PT (Fig. 4c). These significant genes were then thrown into pathway enrichment analysis to uncover the underlying biological functions. We found that the down-regulated genes of expanded CD8 T cells in LNMT were enriched in T cell activation, which also reflects a lower T cell activity in LNMT than in PT (Fig. 4d & Supplementary Fig. 4a).

For CD4 T cells, 460 genes were significantly down-regulated and 21 genes were up-regulated in matched Tconvs of LNMT compared with PT (Fig. 4e). Pathway enrichment analysis revealed that Tconvs in PT are in enriched in the T cell activation pathway and positive regulation of cytokine production (Fig. 4f & Supplementary Fig. 4b). As for Tregs, only 27 genes were down-regulated and 16 genes were up-regulated in the matched Tregs of LNMT compared with PT (Supplementary Fig. 4c). In summary, these data further supported that T cells are less activated in LNMT compared with PT.

## Characteristics of myeloid cells in the breast cancer microenvironment

Myeloid cells play an important role in the TME[23]. We identified 11 clusters of myeloid cells including 4 dendritic cells (DC) subsets (pDC-*LILRA4*, DC-C1-*CD1C*, DC-C2-*CLEC9A*, and DC-C3-*LAMP3*), 6 macrophage subsets (Macro-C1-*APOC1*, Macro-C2-*SLC40A1*, Macro-C3-*VCAN*, Macro-C4-*CXCL11*, Macro-C5-SPP1, and Macro-C6-*CCL3*) and mast cells (Fig. 5a, Supplementary Fig. 5a). The enriched genes of each cell type are shown in Fig. 5b. DC-C1-*CD1C* and DC-C2-*CLEC9A*; the high expression of *DC1C/CLEC10A* and *XCR1/CLEC9A* correspond to cDC1 and cDC2, respectively, and have been well-characterized[24]. DC-C3-*LMAP3*, which was characterized in several recent studies[25, 26], showed a high expression of *CCR7*, chemokines, and costimulatory

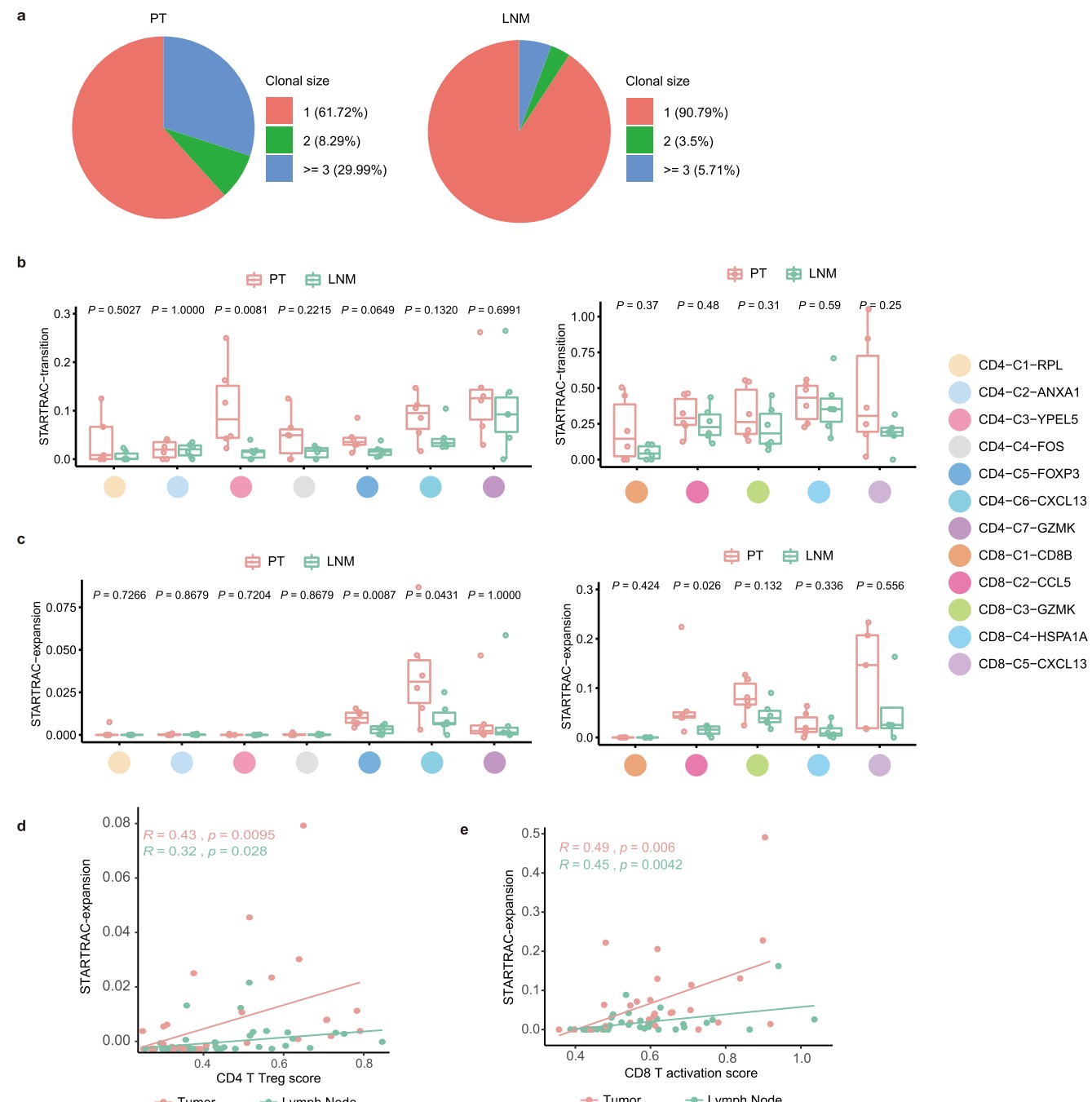

**Fig. 3 | T cells in PT have higher expansion and transition ability than those in LNMT. a** Pie charts showing the fraction of clonal size of shared clonotypes in tumor (left) or LNMT (right). Red represents unique clonotypes, green represents a clonal size of shared clonotypes of 2, and blue represents the clonal size of shared clonotypes of 3 or above. **b** Developmental transition of CD4 T clusters (left) and CD8 T clusters (right) quantified by STARTRAC-transition for each patient (*n* = 6) within each tissue. Two-sided Wilcoxon test. In the box plots, the center line corresponds to the median, box corresponds to the interquartile range (IQR), and whiskers 1.5 × IQR. **c** Clonal expansion of CD4 T clusters (left) and CD8 T clusters (right) quantified by STARTRAC-expansion for each patient (*n* = 6) within each

tissue. Two-sided Wilcoxon test. In the box plots, the center line corresponds to the median, box corresponds to the interquartile range (IQR), and whiskers 1.5 × IQR. **d**, **e** Scatterplot showing the correlation of Tregs score (d) or CD8 activation score (*x*-axis) (**e**) with STARTRAC-expansion (*y*-axis) on CD4 T clusters (**d**) or CD8 T cell cluster (**e**) across different tissues. Tregs score and CD8 activation score were calculated with the average expression of corresponding gene list (Supplement data 3). Each dot represents a corresponding cluster for each patient. Pearson correlation coefficient and linear regression were used. Source data are provided as a Source Data file.

genes, and represented activated DCs (Fig. 5b). An embedding plot revealed differences in DCs across the 2 microenvironments (Supplementary Fig. 5b), with pDC-*LILRA4*, DC-C1-*CD1C*, and DC-C3-*LAMP3* showing a higher preference for LNMTs than for PTs (Fig. 5c). Differential gene analysis showed that DCs in PTs were more enriched in the interferon-stimulated gene (*ISG15*) and had higher levels

of *STAT1* expression (Fig. 5d, Supplementary Fig. 5c), a key transcription for DC differentiation and maturation. Moreover, DC-C3-*LAMP3* in PTs had a higher MHC II gene expression than that in LNMTs (Fig. 5e, f). DC-C3-*LAMP3* in PTs also showed higher enrichment in the glycolysis pathway (Supplementary Fig. 5d), which contributes to DC activation[27]. Because activated DCs play important

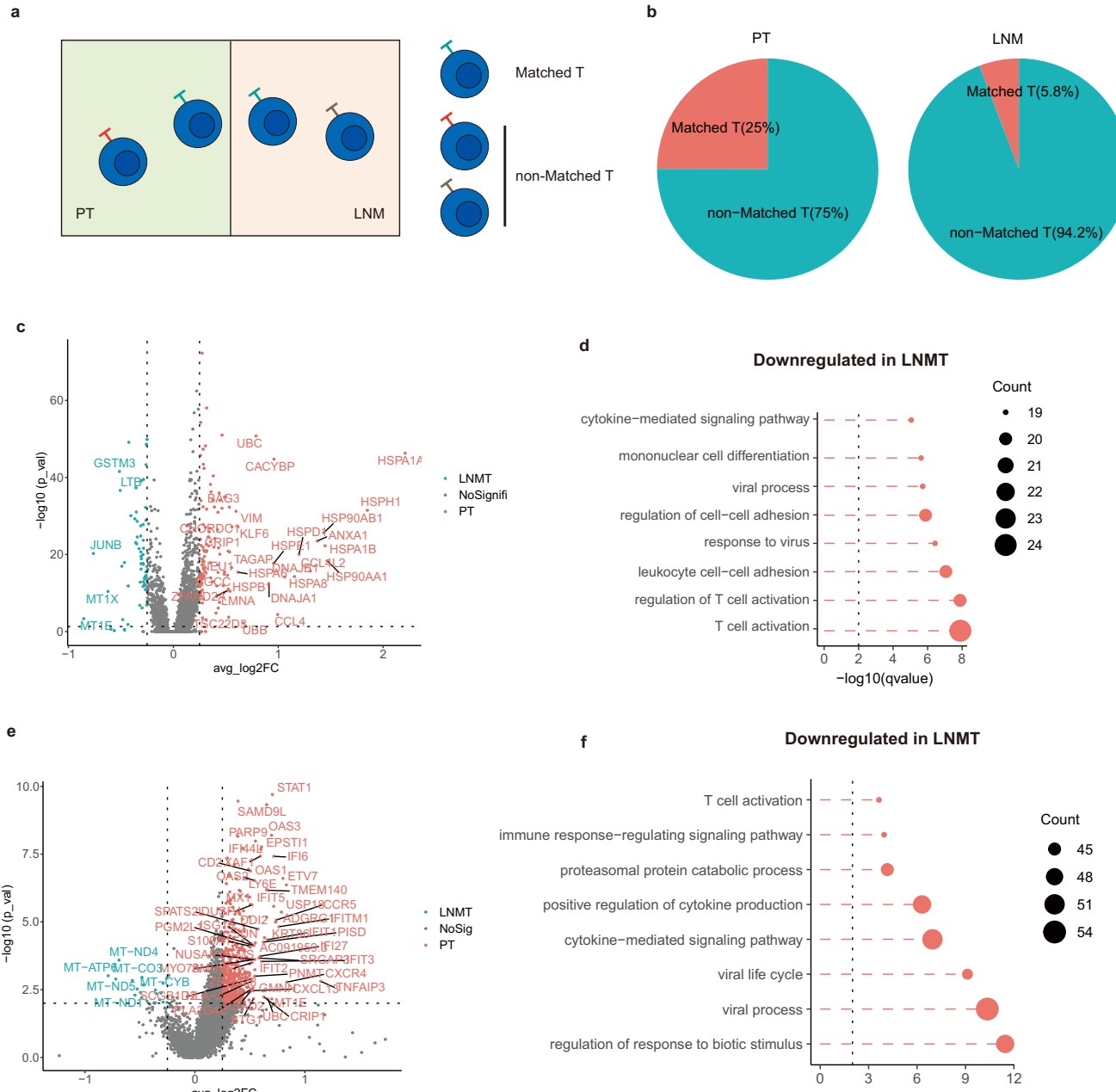

**Fig. 4 | Characteristics of matched T cells in the breast cancer microenvironment. a** Diagram of matched T and nonmatched T cells. **b** Pie charts showing the fraction of matched T and nonmatched T cells in PT (left) or LNMT (right). **c** Volcano plot showing differentially expressed genes between LNMT vs PT for matched CD8 T cells. *P* value < 0.05, log2 (fold change) ≥ 0.25. Statistical testing was performed by a two-sided Wilcoxon test. **d** Top 8 enriched pathways for downregulated genes of matched CD8 T cells in LNMT vs PT. Statistical testing was performed by hypergeometric test. **e** Volcano plot showing differentially expressed genes between LNMT vs PT for matched Tconvs. *P* value < 0.05, log2 (fold change) ≥ 0.25. Statistical testing was performed by a two-sided Wilcoxon test. **f** Top 8 enriched pathways for down-regulated genes in LNMT vs PT. Statistical testing was performed by hypergeometric test. Source data are provided as a Source Data file.

roles in communicating with other immune cells, we then undertook to determine whether DC-C3-*LAMP3* showed a different type of communication with immune cells depending on whether it was in PTs or LNMs. LAMP3⁺ DCs highly expressed *CXCL9* and *CCL19* in PTs and highly expressed *CCL17* and *CCL22* in LNMTs (Fig. 5g). Cell–cell interaction analysis showed that *LAMP3*⁺ DCs in PTs exhibited greater interaction with immune cells through *CXCL9:CXCR3*, *CCL19:CXCR3*, or *CCL19:CCR7* than in LNMTs. However, *LAMP3*⁺ DCs in LNMTs showed a stronger interaction with Tregs through *CCL17:CCR4* and *CCL22:CCR4* (Fig. 5h), suggesting that *LAMP3*⁺ DCs in LNMTs may be more likely to recruit and activate Tregs to enhance

immunosuppression. These data suggested that DCs, especially *LAMP3*⁺ DCs in PTs, were more mature and had a greater T cell priming and activating ability than those in LNMTs.

Subsets with a high expression of *CD68* were defined as macrophages. Tumor-associated macrophages (TAMs) are characterized by enrichment in the TME. We subsequently combined the data of a published study[12] and that of our single-cell analysis to clarify the properties of the tissue-resident macrophage subsets (Supplementary Fig. 5e). Consistent with the previous study, we found that the macrophages in the TME showed a greater degree of diversity and complexity[28]. Macro-C4-*CXCL11* and Macro-C5-SPP*1* were significantly

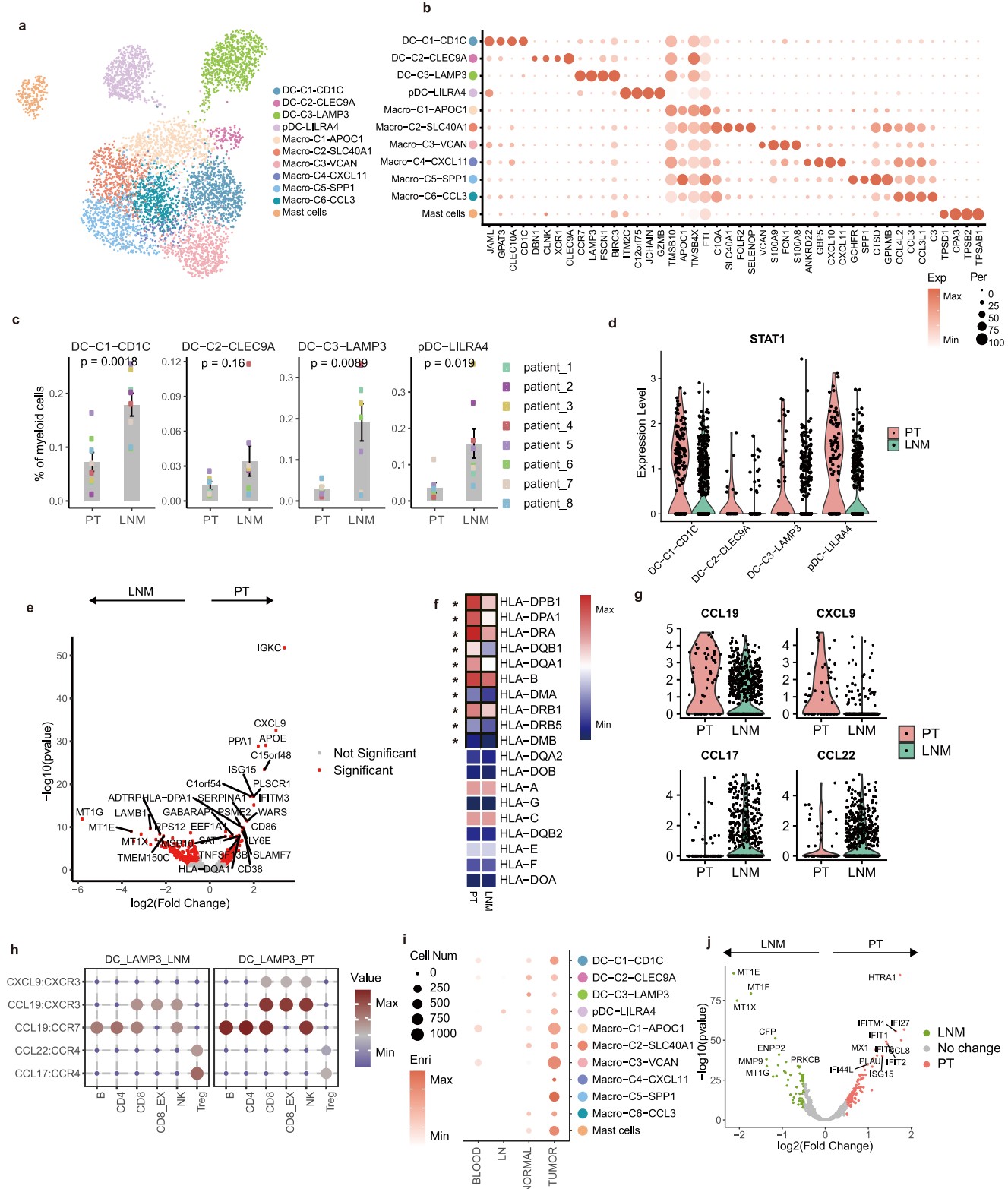

highly enriched in the TME and corresponded to TAMs. Macro-C1-*APOC1*, Macro-C2-*SLC40A1*, and Macro-C6-*CCL3* were defined as tissue-resident macrophages, the levels of which in tumors were comparable to those in adjacent tissues (Fig. 5i). It is well known that M1 and M2 signatures represent the different functions of macrophages in the tumor. In this study, we found there is no difference in the ratio of these 2 types of macrophages in PTs compared to LNMTs

(Supplementary Fig. 5f, g). However, using differential gene analysis, we found that macrophages have a higher expression of *IFI27*, *IFITM1*, and *IFI44L* in PTs than in LNMTs (Fig. 5j), suggesting activation of IFN-γ signaling in PTs. Furthermore, *CCL8*, which has been reported to promote breast metastasis and progression[29], was more highly expressed in PTs than in LNMTs, indicating distinct microenvironments in PT and LNMT that could influence tumor cell survival.

**Fig. 5 | Characteristics of myeloid cells in the breast cancer microenvironment.** **a** UMAP embedding plot of 5992 myeloid cells grouped into 11 clusters. **b** A dot plot showing marker genes across myeloid clusters from the adjacent UMAP embedding plot (a). The dot size indicates the fraction of expressed cells, and the color represents the normalized expression level. **c** Bar plots showing DC clusters between 2 tissues (PT: $n = 8$ samples, LNM: $n = 8$ samples) among myeloid cells. Statistical testing was performed with a two-sided Wilcoxon test. Data are presented as mean values +/− SD. **d** Violin plots showing the STAT1 expression level of DC clusters. **e** A volcano plot showing the differentially expressed genes between PT and LNMT in the DC-C3-*LAMP3* cluster. $P$ value < .05, log2 (fold change) ≥ 0.5. Statistical testing was performed by a two-sided Wilcoxon test. **f** A heatmap showing the expression pattern of MHC II genes between PT and LNMT in the DC-C3-*LAMP3* cluster. The black board pattern indicates the significance of the unpaired 2-sided $t$-test. **g** Violin plots showing the expression of represented chemokines in DC-C3-*LAMP3*. **h** Bubble heatmaps show the interaction strength for representative ligand-receptor pairs between *LAMP3*+ DC and immune cells in different microenvironments; the dot size and color indicate the interaction strength. **i** A dot plot showing tissue enrichment of myeloid clusters based on GSE114727 dataset[12]; the dot size indicates the number of cells, and the color represents the enrichment score. **j** Volcano plot showing the differentially expressed genes between PT and LNMT in macrophages. $P$ value < .05, log2(fold change) ≥ 0.5. Statistical testing was performed by a two-sided Wilcoxon test. The $P$-values were corrected with Benjamini-Hochberg adjustment. Source data are provided as a Source Data file.

## PLA2G2A+ CAFs in HER2+ breast cancer promoted immune infiltration

In our dataset, we identified 3 types of stromal cells: TECs, PVLs, and CAFs[30] (Fig. 6a, Supplementary Fig. 6a). PVLs included 2 subtypes, PVL-C1-*MGS5* and PVL-C2-*MYH11*, both of which exhibited a high expression of actin cytoskeleton gene *ACTA2* and GTPase-activating protein *RGS5* (Supplementary Fig. 6b). CAFs play a pivotal role in the cancer microenvironment and may suppress the immune response and/or promote tumor metastasis[31]. We found there is no difference in the proportion of total CAFs in PTs versus LNMTs (Fig. 1d). According to their differentially coexpressed markers, fibroblasts were further classified into 3 CAF types: myofibroblast-like phenotype (mCAF) including CAF-C1-*POSTN*; and 2 inflammatory property fibroblasts (iCAF), including CAF-C2-*APOD* and CAF-C3-*PLA2G2A* (Fig. 6a). CAF-C1-*POSTN* expressed high levels of *POSTN* and collagen genes (*COL1A2*, *COL3A1*), which have been reported to contribute to extracellular matrix remodeling and stiffness of tumor (Supplementary Fig. 6b)[32]. CAF-C3-*PLA2G2A*, which has been reported in other studies[33], was characterized by a high expression of *OGN*, which encodes a small leucine-rich proteoglycan protein that functions in T-cell recruitment and immune infiltration in tumors[34].

Importantly, we found CAF-C3-*PLA2G2A* at a high frequency in HER2+ tumors, while CAF-C2-*APOD* was enriched in luminal tumors (Fig. 6b, c). Compared with those in luminal breast cancer, HER2+ tumors have a high degree of immune infiltration, and patients with this subtype can benefit from immunotherapy[35]. We thus aimed to determine whether CAF-C3-*PLA2G2A* enrichment in HER2+ tumors had a connection with immune infiltration. Using cell–cell interaction analysis, we found that, compared with the other 2 types of fibroblasts, CAF-C3-*PLA2G2A* showed a much stronger interaction with immune cells, including CD4 and CD8 T cells, DCs, and macrophages (Fig. 6d). Mechanistically, *PLA2G2A*: α4β1 integrin, *OGN*: *HLA-DRB1*, *VSIR*:*CCL4L2*, *FN1*:α4β1 integrin, and *DPP4*:*CCL3L1* interaction complexes were found to be responsible for the association of *PLA2GA*+ CAF with immune cells. *HLA-DRB1*, α4β1 integrin, *CCL4L2*, and *CCL3L1* were reported to be expressed in immune cells[36,37]. In our study, we found that CAF-C3-*PLA2G2A* also exhibited an abundant expression of *OGN*, *VSIR*, *FN1*, and *DPP4*, which can interact with immune cells. In contrast, the gene expression of *OGN*, *VSIR*, *FN1*, and *DPP4* was much lower in CAF-C2-*APOD* (Fig. 6d, e). To further validate the function of *PLA2G2A*+ CAFs in immune cells, we treated monocyte THP1 cells with PLA2G2A protein and found that PLA2G2A could promote the migration of THP1 cells (Fig. 6f). These data suggested the potential of *PLA2GA*+ CAFs to attract immune cells.

Furthermore, we found that *PLA2G2A*, which is exclusively expressed in *PLA2G2A*+ CAFs (Supplementary Fig. 6c), was highly expressed in HER2+ patients and highly correlated with the immune cell marker *CD3E* in the TCGA-BRCA dataset (Fig. 6g). *PLA2G2A* was also positively correlated with the CD45 antigen *PTPRC*, B-cell marker *CD79A*, and CD8 T cell marker *CD8A* in breast cancer patients (Supplementary Fig. 6d). Immunohistochemical (IHC) staining in human breast cancer tissues further revealed that PLA2G2A+ CAFs were more abundant in HER2+ tumors than in luminal tumors (Fig. 6h, i). Immunofluorescence analysis also showed that *PLA2G2A*+ CAFs had a similar spatial distribution to macrophages and CD8 T cells (Fig. 6j), and confirmed the interactions of *PLA2G2A*+ CAFs with macrophages and CD8 T cells at the single-cell level (Fig. 6k). In short, we identified 3 CAF subsets in breast cancer patients and demonstrated enrichment of *PLA2G2A*+ CAFs in HER2+ tumors, and these may be the main microenvironmental factors that determine the immune infiltration in breast cancer.

## Antigen-presentation pathway was down-regulated in malignant cells of metastatic lymph node

Finally, we wanted to characterize the features of malignant epithelial cells. Epithelial cells were divided into malignant epithelial cells and nonmalignant epithelial cells according to their CNVs. Malignant cells and nonmalignant cells were separated according to the distribution of epithelial cell malignancy scores (Supplementary Fig. 7a, b). Heatmaps of CNVs exhibited a considerable degree of divergence among the patients, and similar patterns were observed in the same patients, even in different tissues. Malignant cells from the same patient showed a similar CNV pattern, suggesting that the malignant cells were derived from the same point of origin (Fig. 7a, b).

To further understand the characteristics of malignant cells of lymph node metastasis, we compared transcriptome signatures of the malignant cells between LNMT and PT for patients that have at least 20 cells in each tissue. Thus, 3 patients were excluded and the transcriptome signature of the other 5 patients was calculated (Supplementary data 4). Few significant genes are shared across patients (Supplementary Fig. 7c, d). Interestingly, we found antigen presentation genes, such as *CD74*, *HLA-DRA* and *B2M*, are mostly down-regulated in LNMT compared with PT for patient 8 (Fig. 7c). Similarly, we also found that *HLA-B* and *HLA-C* are down-regulated in LNMT of patient 5 (Supplementary Fig. 7e). To character whether these findings are prevalent in most patients, transcriptome signatures from the 5 patients were used for GSEA enrichment analysis (Supplementary data 5). Significant pathways shared by the 5 patients were analyzed and the normalized enrichment score (NES) for each pathway was averaged. The pathways were then ranked according to the numbers of shared patients and the mean of NES. From the top 10 enriched pathways, 4 pathways are related to antigen presentation and these pathways are enriched in 4 of 5 patients (Fig. 7d, e, Supplementary data 6).

It is interesting to ask whether the down-regulated antigen presentation pathway is related to CNV clones of malignant cells. Then we clustered malignant cells according to their CNV similarity for each patient and then compared the DEG of each CNV clone in PT vs LNMT (Fig. 7f). We found that malignant cells belonging to different CNV clusters in patient 8 have no difference in metastasis ability (Fig. 7g). And the malignant cells in different CNV clusters of patient 8 are also enriched in the antigen presentation pathway (Supplementary Fig. 7f, g). Antigen presentation genes can be mainly divided into MHC I and MHC II class molecules. We compared the two types of antigen presentation in PT vs LNMT across the different CNV

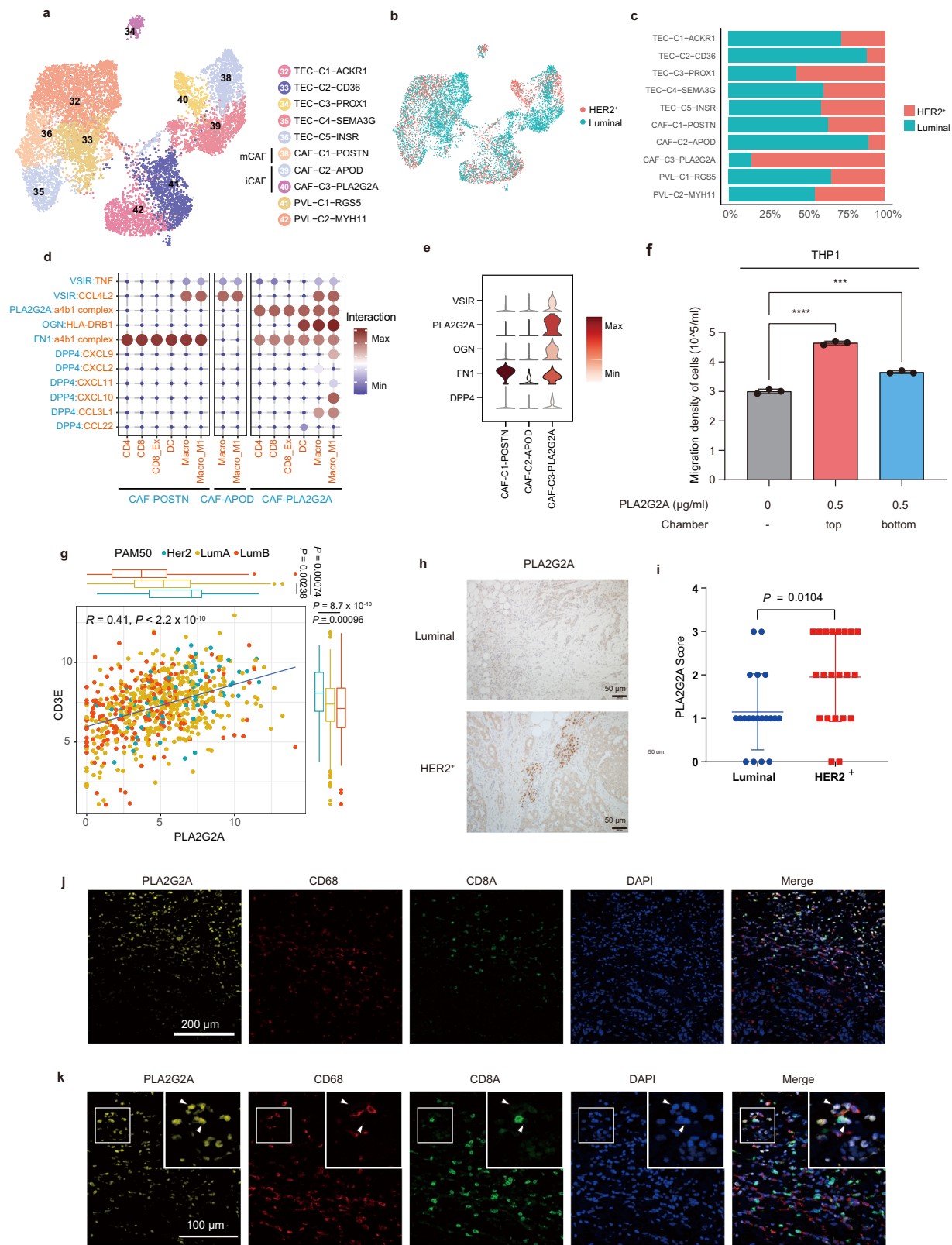

clusters and found that both MHC I and MHC II class molecules are down-regulated in LNMT across different CNV clusters in patient 8 (Fig. 7h). MHC I genes but not MHC II genes are down-regulated in LNMT of Patient 5 (Supplementary Fig. 7h–j). These findings indicate that malignant cells migrated to LNMT may render lower antigen presentation genes, resulting in an immune evasive mechanism, which

provide insight into the characterization of malignant cell metastasis in breast cancer.

## Discussion

Single-cell sequencing is a powerful tool for analyzing the TME[6]. Several breast cancer studies have used single-cell sequencing[12,38–40] to

**Fig. 6 | PLA2G2A⁺ CAFs in HER2⁺ breast cancer promoted immune infiltration.**
**a**, **b** A UMAP embedding plot of 10,049 fibroblasts and vascular endothelial cells grouped into 10 clusters. Cells were color-coded according to cluster (**a**) or subtypes (**b**). **c** The proportion of subtypes within 10 subsets in fibroblasts and vascular endothelial cells; the color represents the tissue type. The x-axis is the fraction of tissues, and the y-axis is the major cell type. **d** A bubble heatmap showing the interaction strength for representative ligand-receptor pairs between CAFs and immune cells; the dot size and color indicate the interaction strength. **e** Violin plots showing the expression of proteins that interacted with immune cells of **d**. **f** Transwell assays were used to measure cell migration. PLA2G2A protein (0.5 μg/ml) was respectively applied to the top chamber and the bottom chamber, and then cells were incubated at 37 °C in 5% CO$_2$ for 4 h. Statistical testing was performed by two-sided t-test. (n = 3 biological replications; ***P = 0.000706, ****P = 0.000011). Data are presented as mean values+/− SD. **g** The correlation between PLA2G2A and CD3E of HER2⁺ and luminal patients in the TCGA-BRCA datasets (HER2⁺: n = 67 samples, LumA: n = 421 samples, LumB: n = 192 samples). The correlation coefficient and P-value were calculated with two-sided Pearson rank correlation. In the box plots, the center line corresponds to the median, box corresponds to the interquartile range (IQR), and whiskers 1.5 × IQR. Statistical testing was performed with a two-sided Wilcoxon test. **h** IHC staining showing the PLA2G2A protein levels in representative luminal (left) or HER2⁺ (right) patients. **i** PLA2G2A score of IHC either in luminal (n = 20) or HER2⁺ (n = 21) breast cancer patients. Two-sided Wilcoxon test. Data are presented as mean values+/− SD. **j**, **k** Representative images of a HER2⁺ patient stained by multicolored IHC; yellow represents PLA2G2A⁺ CAF, red represents macrophages, and the green represents CD8 T cells. Original magnification, 40x (**j**), scale bar = 200 μm or 100x (**k**), scale bar = 100 μm. (n = 3 independent replications). Source data are provided as a Source Data file.

analyze the differences of immune cells among breast cancer in situ, adjacent cancer, blood, and normal LNs. However, an analysis of the metastatic LN microenvironment in breast cancer at the single-cell level has not yet been reported. We thus systematically revealed the characteristics and differences in the microenvironment between LNMTs and paired PTs in breast cancer.

The TME is shaped into an environment with characteristics, such as immunosuppression, that are more conducive to tumor growth. For example, we found that the TME induced the transformation of CXCL11⁺ macrophages with the manifestation of M1-like macrophages into SPP1⁺ macrophages with a higher M2 signature, promoting cancer progression. However, compared with the TME in situ, the microenvironment of lymphatic metastasis exhibited stronger immunosuppression. Lymphatic vessels play an important role in tumor immunity, aiding the antigen presentation of DCs and the activation of T cells in the TME[41]. The particularity of lymphatic vessel structure and the high expression of CXCR4 in breast tissue cells may be an important reason for metastasis to and colonization of LNs[42]. Our study investigated this mechanism by performing a single-cell analysis and by characterizing the microenvironment of LNMTs. We found that the overall activity of T cells in metastatic LNs was suppressed more than that in the PTs. This conclusion is supported by several lines of evidence: (1) from the transcriptional level, CD8 T cells showed weaker activity and cytotoxic effect in the LNMT microenvironment compared with those in PTs. Although it is well known that CD8 T cells in PTs have higher activation than those in LNMTs, we identified the phenotype and characteristics of CD8 T cells within the two tissues in detail, such as the subpopulation and development path of CD8 T cells in PT vs LNMT, which can be well characterized in single-cell sequencing. (2) CD4⁺CXCL13⁺ cells were reprogramed to an exhausted state and expressed lower levels of IFNG, CXCL13, and genes that inhibit tumor growth in the LNMT microenvironment. However, the function of the 5 clusters of CD4⁺CXCL13⁺ cells needs to be further verified by experiments. (3) The frequency of CD8⁺CXCL13⁺ was remarkably decreased in LNMT. And (4) compared to those in PTs, MT cells in LNMTs exhibited functional inhibition. Collectively, these findings indicate that immune cells in metastatic LNs are reprogrammed in several ways. Lymphatic vessels express several genes that inhibit T cell activity, such as programmed death-ligand 1 (PD-L1), and may mediate immune tolerance[43,44], a fact which may be the key to explaining our findings. Furthermore, very interestingly, we found that malignant cells migrated to LNMT downregulate their antigen presentation pathway, which results in the defect of presenting tumor-related epitope to adaptive immune cells. Loss of antigen presentation has been studied in various tumors and was the common way for cancer cells to escape from immune surveillance. Taken together, these findings may help us understand why the tumor cells are easy to survive in LN.

Another interesting finding in this study is that activated LAMP3⁺ DCs showed higher enrichment in LNMTs than in PTs and strongly interacted with Tregs through CCL17 or CCL22, and this may partly account for the suppressed activity of T cells in LNMTs. A recent study revealed that LAMP3⁺ DCs in pancreatic adenocarcinoma might promote immune tolerance through interacting with tumor-infiltrating Tregs, supporting our conclusions[45]. Given the higher proportion of naïve state T cells in LNMTs compared to in PTs, we speculate that immunotherapy drugs may have very limited antitumor effects on the microenvironment of LNMTs, and this may have ramifications for the clinical treatment of breast cancer with LNMTs.

Immune infiltration in tumors is currently an area of research focus, and findings in this area may have important clinical application[46]. Immune infiltration is related to different subtypes of breast cancer. Compared with patients of other subtypes, patients with triple-negative and HER2⁺ breast cancer exhibit stronger immune infiltration and can benefit from immunotherapy[35,47]. Although clinical and pathological characteristics have been found to be related to immune infiltration, the current understanding of immune cell infiltration is extremely limited, and the mechanisms underlying the varying degrees of immune infiltration in patients with different subtypes are still unclear. CAFs are important components in the TME by their regulation of immune cell activity and promotion of tumor progression and metastasis. In this study, we found a type of PLA2G2A⁺ CAFs that was enriched in HER2⁺ breast cancer and showed high expression levels of genes that can interact with immune cells. PLA2G2A was reported to promote the proliferation of monocytic cells through interacting with αvβ3 and α4β1 integrin[48]. A previous report also found that PLA2G2A is overexpressed in some fibroblast subtypes[49]. We verified in the TCGA-BRCA database that there was a significantly positive correlation between PLA2G2A⁺ CAF markers and immune cell markers. Furthermore, in our single-cell dataset, PLA2G2A⁺ CAFs were observed to interact with various immune cells, especially macrophages, which suggests that PLA2G2A⁺ CAFs may be a factor in tumor immune infiltration, promoting tumorigenesis and tumor development. Another interesting observation is that PLA2G2A⁺ CAFs were present in both the PT and the LNMT of HER2⁺ patients; however, they were not present in the LNMT of patients with luminal breast cancer. Therefore, PLA2G2A⁺ CAFs may be recruited and dominated by HER2⁺ tumor cells to migrate or differentiate in the process of tumor development.

In summary, we demonstrated that the immune cells in LNMTs are less active than those in PTs and clarified the mechanism underlying this difference. Further mechanistic investigations to determine precisely why tumor cells can more easily colonize and proliferate in LNs are thus warranted. Our study identified the unique features of the LNMT and PT microenvironments, the knowledge of which can aid in developing individualized therapy that targets these microenvironments in patients with breast cancer.

## Methods

The study was conducted in accordance with the Declaration of Harbin Medical University complying with all relevant ethical regulations.

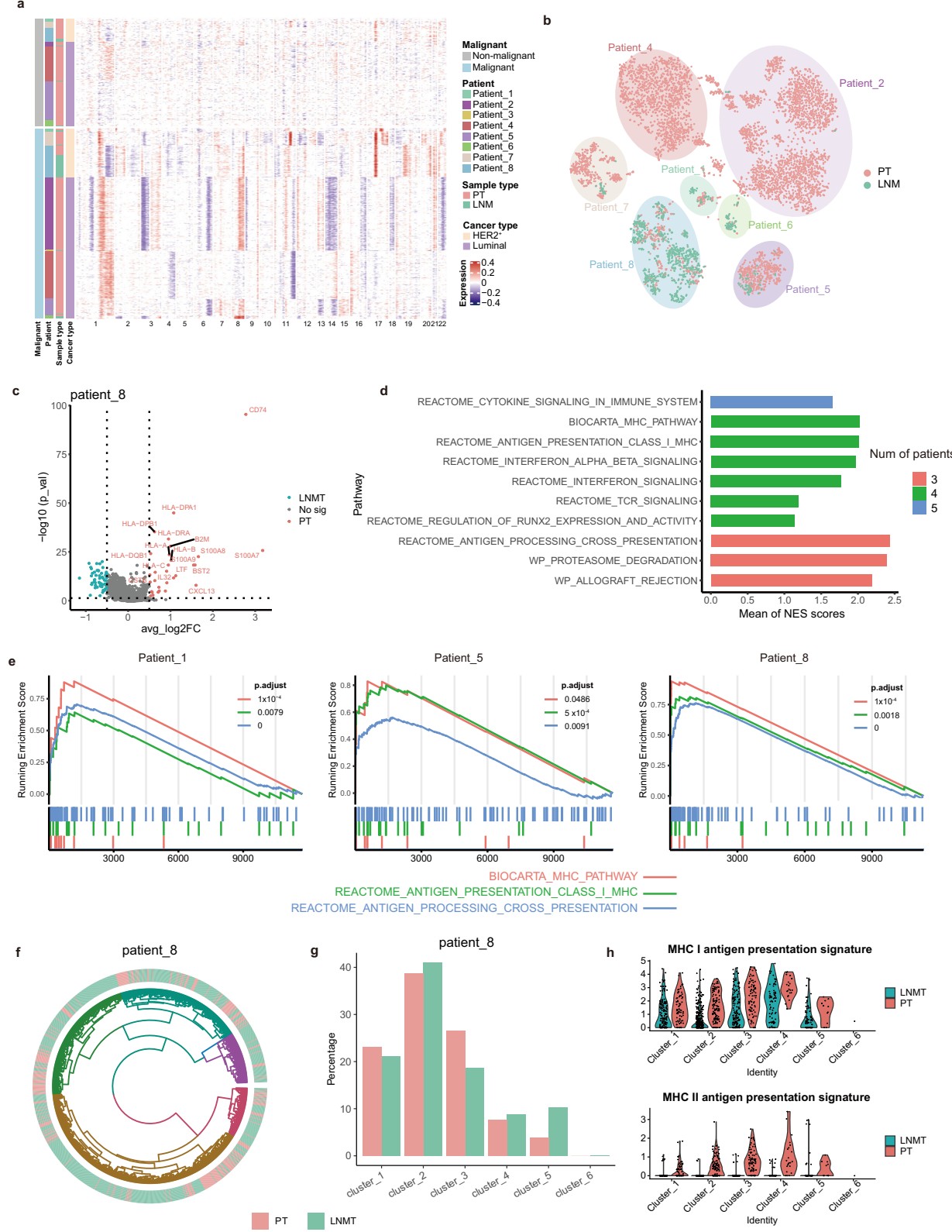

Written informed consent was obtained from all participants. No compensation was provided for the study participants.

## Tissue acquisition

Eight treatment-naïve female patients with a pathological diagnosis of invasive ductal carcinoma of the breast with LNMT were enrolled at Harbin Medical Hospital. Their ages ranged from 47 to 66 years, with the median age being 56 years. PT and paired LNMT tissues were surgically resected from each patient. Among the patients, 5 cases were luminal subtypes and 3 cases were the HER2-overexpressing subtype.

This study was approved by the Research and Ethical Committee of Harbin Medical University Cancer Hospital (IRB:KY2019-08). Written informed consent was obtained from all participants in the study.

**Fig. 7 | Antigen-presentation pathway was downregulated in malignant cells of metastatic lymph node. a** A heatmap showing large-scale CNVs of epithelial cells (rows) from paired tumor tissues in 8 LNMT patients. CNVs in red indicate amplifications, while those in blue indicate deletions. CNVs were identified by inferCNV. **b** t-SNE plot of malignant epithelial cells identified by inferCNV malignant score. **c** Volcano plot showing the differentially expressed genes between PT and LNMT in malignant cells of patient 8. *P* value < 0.05, log2 (fold change) ≥ 0.5. Statistical testing was performed by a two-sided Wilcoxon test. **d** Barplot showing top 10 normalized enrichment score (NES) of shared significant enriched pathways across patients. Genes ranks calculated by PT vs LNMT in malignant cells for each patient were used for NES calculating by GSEA analysis. Color represents the numbers of patients which are significant in same pathways. **e** GSEA analysis showed that genes rank in PT vs LNMT of represented patients are enriched in antigen presentation pathway. Left represent high gene rank in PT. Statistical testing was performed by permutation test. The *P*-values were corrected with Benjamini-Hochberg adjustment. **f** Circos plots showing 6 CNV clusters of malignant epithelial cells according to CNVs similarity for patient 8. Cells were colored by tissue. **g** Bar plot showing the frequency of 6 CNV clusters in patient 8. **h** Violin plots showing the MHC I antigen presentation signature (Up) and MHC II antigen presentation signature (Down) of 6 CNV clusters from patient 8. Source data are provided as a Source Data file.

## Cell preparation

PT or paired LNMT tissues were separated and digested into single-cell suspensions. The 10x Genomics Cell Preparation Guide describes best practices and general protocols for washing, counting, and concentrating cells from both abundant (>100,000 total cells) and limited cell suspensions (<100,000 total cells) in preparation for use in 10x Genomics Single Cell Protocols.

Formalin-fixed, paraffin-embedded (FFPE) samples passing the RNA quality control were used to prepare for spatial transcriptomic construction and sequencing. The Visium Spatial Gene Expression Slide & Reagent Kit (10x Genomics) was used to construct sequencing libraries according to the Visium Spatial Gene Expression User Guide (CG000239, 10x Genomics).

## Library construction and sequencing

The cell suspension was loaded into Chromium microfluidic chips with 5′ v.1.1 chemistry and barcoded with a 10× Chromium Controller (10x Genomics). RNA from the barcoded cells was subsequently reverse-transcribed, and sequencing libraries were constructed with reagents from a Chromium Single Cell 5′ v1.1 Reagent Kit (10x Genomics) according to the manufacturer's instructions. Sequencing was performed with an Illumina NovaSeq 6000 PE150 system, depending on the experiment and following the manufacturer's instructions.

Spatial transcriptomic sequencing was performed with a NovaSeq PE150 platform according to the manufacturer's instructions (Illumina) at an average depth of 300 million read-pairs per sample.

## Multicolor immunohistochemistry

Fresh tissues obtained from the patients were embedded in paraffin. The paraffin-embedded tissues were cut into 5-um–thick sections on a glass slide. The sections were infiltrated with fresh xylene 3 times for 10 min each time before being soaked in 100% ethanol, 95% ethanol, and 75% ethanol, in that order, once for 5 min. The sections were then soaked in sterile water 3 times, for 1 min each time, to remove the paraffin. The deparaffinized slides were exposed to antigen in a 100 °C water bath with antigen retrieval solution for 20 min. Next, the slides were blocked with 10% goat serum for 10 minutes. After the removal of the blocking solution, the slides were incubated with the first primary antibody at room temperature for 1 h and then with the secondary antibody at room temperature for 10 min. Finally, the slides were incubated with fluorescent staining amplification solution (Absin multicolor immunohistochemistry kit) for 10 min at room temperature. The second and third primary antibodies were stained following the same steps as those for the first antibody, including antigen retrieval, and incubation with primary antibody, secondary antibody, and fluorescent amplification solution. Primary antibodies used for multicolor immunohistochemistry were rabbit anti-human CD8A (Abcam, catalog: ab17147, clone id: C8/144B, 1:200 dilution), rabbit anti-human CD68 (Abcam, catalog: ab213363, clone id: EPR20545, 1:200 dilution), and anti-human PLA2G2A (Invitrogen, catalog: PA-102403, 1:200 dilution).

## Single-cell RNA-sequencing data processing

Droplet-based sequencing data were aligned and quantified with the Cell Ranger Software Suite (version 3.1.0, 10x Genomics) using the GRCh38 human reference genome (official Cell Ranger reference, version 3.0.0). To obtain high-quality cells, every sample underwent filtering as follows: (1) cells were filtered if the number of detected genes (log10 scale) was below the medians of all cells minus 3 × the median absolute deviation; (2) cells were filtered if the proportion of mitochondrial genes was higher than the median of all cells plus 3 × the median absolute deviation; and (3) cells were filtered if their unique molecular identifier (UMI) counts were lower than 300.

For spatial transcriptomic sequencing, FASTQ files and histology images were processed by Space Ranger (version spaceranger-1.2.0, 10x Genomics) software with default parameters. The filtered gene-spots matrix and the fiducial-aligned low-resolution image were used for down-streaming data analyses (Seurat).

## Doublet detection

To remove doublets in single-cell RNA sequencing data, cell doublets were identified using the Scrublet package[50]. Briefly, for each sample, the cell count matrix was fed to Scrublet. Then, the "scrub_doublets" function was applied to simulate doublets, the doublet scores were calculated, and doublet calling was performed. To further reduce the false-negative rate in the Scrublet analysis, we over clustered the remaining cells and calculated the average doublet score within each cluster. We removed any clusters that had an average doublet score of more than 0.6 or more than 1 known cell marker (i.e., *CD3D* for T cells and *CD79A* for B cells).

## Estimation and removal of contaminated messenger RNA

We observed ambient messenger RNA (mRNA) effects, which are ubiquitous in droplet-based single-cell RNA-sequencing (RNA-seq) experiments. The possible reason for this is that free mRNA released from dying cells and single cells were simultaneously captured by beads in droplets. *SCGB2A2* is expressed almost exclusively in the normal breast epithelium and human breast cancer cells; however, in our dataset, we found other cell types. such as immune cells and CAFs, also expressed *SCGB2A2*. We stained *SCGB2A2* and immune cell markers to confirm the absence of these markers. We then used SoupX software to remove or reduce the ambient mRNA effects[51]. Briefly, we regrouped all cells ("sc.tl.leiden" function from the scanpy package; resolution = 0.8) to obtain a rough cluster classification. Raw count matrices with defined clusters were fed into the "SoupChannel" and "setClusters" functions. To estimate the contamination fraction of the dataset, we manually defined the 3 gene sets with the strongest ambient effect: immunoglobulin (IG) genes, human leukocyte antigen (HLA) genes, and breast epithelium genes (*SCGB2A2* and *KRT19*). These gene lists were fed into the "estimateNonExpressingCells" function and the "calculateContaminationFraction" function to calculate the contamination fraction for each cell. Finally, the original count matrix was automatically corrected using the "adjustCounts" function, and the adjusted matrix was further used for downstream biological analysis.

## Dimension reduction and annotation of single-cell RNA-seq data

To integrate and visualize data, we used the Scanpy python toolkit (version 1.4.1) to analyze our single-cell dataset. Briefly, we performed dimension reduction steps including normalization, logarithmic transformation, highly variable gene calling, data scaling, and PCA calling. To remove the batch effect, data were processed using batch-balanced k-nearest neighbors (BBKNN). BBKNN modifies the neighborhood construction step to produce a graph that is balanced across all batches of the data. This approach treats the neighbor network as the primary representation of the data. For each cell, the BBKNN graph is constructed by finding the k-nearest neighbors for each cell in each user-defined batch independently. As for our data, we implemented BBKNN from the scanpy package by using the "bbknn function"[52], and the parameters were set as follows: batch_key = "patient", n_pcs = 35. A batch-corrected neighbor graph was used to find clusters using the Leiden community detection algorithm. To reasonably cluster cells and find their biological markers, the following steps were performed. First, we changed the Leiden resolution parameter from 0.6 to 2 by 0.2 to obtain a collection of cell classifications. Then, we compared the UMAP embedding plot colored by canonical markers (*PTPRC*, *CD3D*, *CD8A*, *CD4*, *CD79A*, *LYZ*, *PLVAP*, *ACTA2*, and *KRT19*) with the UMAP embedding plot colored by the clusters output by different Leiden resolutions to find the smallest suitable Leiden resolution to distinguish canonical markers. Clusters with the same canonical markers were merged. In the first round of clustering, we identified 6 major cell types including immune T and natural killer (NK) cells, epithelial cells, B cells, myeloid cells, CAFs, and tumor endothelial cells (TECs). We observed that immune cell types, including CD4, CD8, and NK cells could not be distinguished well using the Leiden-based classification in the first-round clustering. The reasons for this include the cell types detected by 10x Genomics having limited features and similar transcriptome profiles, and dimension reduction preserving the difference between major cell types but losing some information about major cell types. In the second round of clustering, immune T and NK cells, epithelial cells, myeloid cells, CAFs, and TECs labeled in the first round were further divided into subsets and reclustered into more detailed subclusters. To identify the specific markers of each cluster, differentially expressed genes were identified using the "FindAllMarkers" function of the Seurat R package (v. 3.1.5). Clusters were annotated based on the expression of known marker genes or the most highly expressed genes (Supplementary data 1). The third round of clustering was performed on the annotated clusters in which we were interested, including the group of *FOXP3*⁺ expressed *CD8A*⁺ markers that were distinguished when we reanalyzed CD4-C5-*FOXP3* and re-clustered CD4-C6-*CXCL13* to find heterogeneity and continuous biological states. The second and third rounds of clustering were performed following the same steps as those in the first round, starting from the adjusted count matrix and including normalization, logarithmic transformation, variable gene calling, scaling, PCA calculation, and batch correlation.

## Estimating the cellular composition of each sample

To compare the relative preference of each cell type in different classifications (e.g., PT vs. LNMT and HER2⁺ vs. luminal), we made a double table of the number of cells according to the corresponding classification. To reduce the sample size effects, we calculated sample size scaled proportion as follows: number of cells of a specific type in a category/total number of cells in the category. The category-normalized numbers were used to calculate the proportions of categories in a specific cell type.

To calculate the cellular composition in a specific patient, we defined broad cell types (e.g., B, CD4, CD8, DC, macrophage, CAF, and epithelial cell). We calculated the number of cells of a specific type as defined above and divided this number by the total numbers of cells from a specific patient. As the tumor cell suspensions were thoroughly mixed and captured without bias, this ratio reflected the natural cellular composition within the tumor.

## Integration of spatial transcriptomic with single-cell RNA-seq with Seurat

The Seurat package was used to perform gene expression normalization, dimensionality reduction, spot clustering, and differential expression analysis. To integrate the data, single-cell RNA-seq transcriptome and spatial transcriptome were preprocessed by the "SCTransform" function and PCA analysis. Then "FindTransferAnchors" and "TransferData" functions were used to measure each cluster score for each spot.

## Comparison of myeloid cells in different tissues in other datasets

We downloaded count matrix files (GSE114727)[12] from the Gene Expression Omnibus (GEO) database and mapped labels and embeddings from reference data to the new datasets. In brief, raw count matrix data from the new dataset were imported into scanpy and subjected to preprocessing, including filtering, normalization, and logarithmic transformation. We then used the "ingest" function in scanpy to integrate embeddings and annotate the new datasets through projection onto a PCA that was fit our reference data. The mapping labels from the new datasets were output for enrichment analysis.

## Estimation of CNV and determination of malignant cells

We inferred the CNV of epithelial cells using transcriptomic profiles, to determine the malignant epithelial cells[7]. CNV was estimated based on 2 major steps: initial CNV ($CNV_i$) calculation and final CNV ($CNV_f$) estimation. Genes were first sorted according to their genomic location at each chromosome, and then the $CNV_i$ was derived by applying a sliding window of 100 genes to calculate the average relative expression values within each chromosome. In this way, gene-specific patterns could be eliminated, and the derived profiles (i.e., moving average) largely reflected the CNV. We also restricted the relative expression values to [−3, 3] (values beyond the bounds were replaced with bound values) to avoid any genes with extreme expression influencing the moving average. We defined the CNV score of every single cell as the sum of the square of the $CNV_f$ across all windows. The malignancy score of each single-cell was defined as the mean of the square of the $CNV_f$ minus 1 across all windows. The smooth distribution curve of the malignancy score was fit using bimodal methods to estimate the threshold for malignancy. This function was provided by the "getBimodalThres" function in the scCancer package[53]. Cells with a malignancy score that exceeded the malignant threshold were determined to be malignant epithelial cells.

## Cell–cell interaction analysis

Cell-cell interactions were analyzed using the cellphoneDB python package[54]. To reduce the computational burden and represent different cell types, we downsampled the dataset by randomly sampling 1000 cells of each cell type. The strength of interactions was computed based on the expression of a receptor by 1 cell type and a ligand by another cell type. Only receptors and ligands expressed by more than 30% of the cells in a specific cluster were considered. The cluster labels of all cells were randomly permuted 1000 times to calculate the *P* value for the likelihood of paired interactions. Only paired interactions with a *P* value of less than 0.05 were considered. To obtain different paired interactions between different types of CAFs and other cell types, we calculated the differentially expressed genes using the "FindAllMarkers" function. Only genes meeting the criteria of LogFC threshold > 0.5 and min.pct > 0.25 were used to filter paired interactions.

### Diffusion map–based pseudotime trajectory inference

We implemented a diffusion map on cell types that had the same lineage, such as CD8 T cells. A diffusion map function was implemented in scanpy packages. The BBKNN batch-corrected expression matrix was used for calculating the neighbor matrix with the following parameters: n_neighbors = 10, n_pcs = 15, and method = "gauss". The neighbor matrix was then used to calculate the diffusion map. We found that changing the number of neighbors did not impact the relative position of cell types in the diffusion map.

### Monocle 2–based pseudotime trajectory inference

We constructed the single-cell trajectory of epithelial tumors by using a reversed graph embedding method implemented in the R Monocle 2 package (v. 2.6.3)[55]. To increase the efficiency of the operation, we randomly selected 500 cells annotated as epithelial cells from each sample; for sample sizes of fewer than 500 cells, all the cells were taken. We integrated the expression matrix of epithelial cells using BBKNN, to neutralize patient-specific effects, including different patients and disease subtypes. After this, cell clusters were determined by a Leiden function with a resolution of 0.8 in batched-removed epithelial cells. We compared each cell cluster with other clusters using the "FindAllMarker" function in the Seurat package to determine the batch-removed differential expressed genes, and the top 20 differential expressed genes per cluster were used to order cells in Monocle to construct the epithelial cell DDRtree trajectory plot[56]. The signature of each state was calculated based on differentially expressed genes over the other states. To compare breast cancer epithelial cells with normal epithelial cells, we first obtained signature genes of normal epithelial developmental states from normal epithelial datasets. Then the average of the signatures was calculated for breast cancer epithelial cells.

### T-cell receptor analysis

Alignment and quantification of 10× VDJ sequencing data were performed with the Cell Ranger software using the GRCh38 human VDJ reference genome (official Cell Ranger reference, v. 3.1.0). VDJ sequencing information was extracted from the output file using the Scripy python package (v. 0.3). We used the "chain_pairing" function to summarize TCR compositions. Cells with the same α or β chains were defined as clonotypes. According to their unique α and β chains, TCR chains can be classified into 7 types: single pair, orphan beta, orphan alpha, extra alpha, extra beta, 2 full chains, and multichain. Only the single pair type was used in the downstream analysis. To integrate transcriptome data, we included cells annotated as CD4 or CD8 T cells to visualize our TCR data on embeddings.

We used STARTRAC packages[20] to analyze the behavior of T cells. To obtain good quality data, we excluded patient 2 and patient 4, who had low T-cell capture rates, from the downstream analysis. STARTAC-expansion and STARTAC-transition were used to separately analyze T cells in PTs and LNMTs.

STARTRAC-expansion is usually used in the standard TCR clonality measurement but was specifically applied to different T cell clusters in our analyses. Normalized Shannon entropy was used to calculate the evenness of the TCR repertoire of the given T cell cluster and then define the STARTRAC-expansion index as 1-evenness. STARTRAC-expansion ranged from 0 to 1, with 0 representing no clonal expansion for each clonotype and 1 representing a cluster composed of only 1 clonally expanded clonotype. A high STARTRAC-expansion indicated high clonality.

STARTRAC-transition was used to quantify the extent of state transition of each clonotype within a given cluster. The STARTRAC-transition index at the cluster level was defined as the weighted average of all TCR clonotype state transition indices contained in the cluster.

### Cell culture

THP-1 (human acute monocytic leukemia cell line, 1101HUM-PUMC000057) was purchased from Cell Resource Center of Peking Union Medical College (Beijing, China). THP-1 was authenticated using short tandem repeat analysis. No mycoplasma contamination was detected. THP-1 cells were cultured in RPMI-1640 medium supplemented with 10% fetal bovine serum, penicillin (100 μg/ml) and streptomycin (100 μg/ml). Cell lines were incubated in a humidified atmosphere of 5% $CO_2$ at 37 °C.

### Transwell assay

$2 \times 10^6$ THP1 cells were added into the top chamber of 24-well tissue culture inserts (Costar). PLA2G2A (0.5 μg/ml, 11187-H08H, Sino Biological) was respectively applied to the top chamber, in RPMI-1640, and the bottom chamber, in RPMI-1640 containing 20% serum. After incubation at 37 °C in 5% $CO_2$ for 4 h, cells in the bottom chamber were collected and counted.

### The Cancer Genome Atlas data analysis

The Cancer Genome Atlas Breast Invasive Carcinoma (TCGA-BRCA) datasets were used to analyze the correlations of selected genes with patient survival. Gene expression as well as clinical and survival data were downloaded from UCSC Xena (http://xena.ucsc.edu/). The signature scores of the TCGA-BRCA patients were calculated as the mean expression of genes in the signature. The signature scores were grouped into high and low expression groups by the 55th and 45th quantile values, respectively. We used the survival packages to calculate the impact of genes or signatures on patient survival and plotted Kaplan–Meier survival curves using ggsurvplot in R. For correlation analysis, gene or signature scores were calculated by applying Spearman's correlation coefficient using the cor function in R.

### Reporting summary

Further information on research design is available in the Nature Portfolio Reporting Summary linked to this article.

## Data availability

The raw data and processed data of single-cell RNA-sequencing data and single-cell VDJ-sequencing data have been deposited in the Gene Expression Omnibus (GEO) database under accession code GSE167036. The raw data and processed data of spatial transcriptomic data generated in this study have been deposited in the Gene Expression Omnibus (GEO) database under accession code GSE190811. The publicly available single cell dataset used in this study are available from the Gene Expression Omnibus (accession numbers GSE114727[12]). Source data are provided in this paper as a Source data file. The remaining data are available within the Article, Supplementary Information, or Source Data file. Source data are provided with this paper.

## Code availability

Code can be found on GitHub: [https://github.com/bio-liucheng/brca-singlecell] and Zenodo and the corresponding DOI is as follows: https://doi.org/10.5281/zenodo.7212881[57].

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

## Acknowledgements

This work was supported by grants from the Ministry of Science and Technology of the People's Republic of China (2021YFC2501000, H.Z.); National Natural Science Foundation of China (81730071 H.Z., 81972609 H.Z., 82230094 H.Z., 82073240 X.W., 82072903 T.L., 82002811, M.Y.); PKU2020LCXQ007 and PKU2021LCXQ023 to H.Z, and the Beijing Natural Science Foundation (7202080 X.W.). This study was also supported by Clinical Medicine Plus X-Young Scholars Project of Peking University (X.W.). The authors appreciate the academic support from the AME Breast Cancer Collaborative Group.

## Author contributions

The study was designed by H.Z., X.W., and C.L.; Tissue acquisition was coordinated by T.L and M.Y. Tissue preparation and sequencing were performed and supervised by T.L. and Z.LI.; Bioinformatic analysis was conducted by C.L. and L.Z., and supervised by H.Z. and X.W.; IHC experiments were designed by X.W. and performed by J.Z.; HE staining was performed by M.X. The figures were drafted by C.L with help of the rest of the authors. The manuscript was written by X.W. and C.L., reviewed by H.Z. and Z.LI. The project was supervised by Z.LI., X.W. and H.Z.

## Competing interests

The authors declare no competing interests.
