## [Peer Review File · Nature Communications]

Single Cell Profiling of Primary and Paired Metastatic Lymph Node Tumors in Breast Cancer PatientsREVIEWER COMMENTS

Reviewer #2, expert in breast cancer microenvironment and single cell RNA-seq (Remarks to the Author):

This manuscript reports on the analysis by scRNA-Seq of 8 primary breast cancers (PT) and paired metastatic lymph nodes (LNM). Overall the study is well collected and the data is of good quality, but the analysis is unfocussed and incomplete. Lymph node metastasis (LNM) is a common occurrence in breast cancer, but little data exists comparing the cells found in LNM and PT. This will therefore be a useful resource for those in the field.

The authors take a broad 'landscape' approach to describe the immune, stromal and epithelial components of these datasets.

Much of the manuscript focusses on immune cells, using both scRNA-Seq and TCR sequencing data. The clustering is well performed and consistent with other literature. However, the manuscript fails to reach conclusive findings a number of times, and claims are often not sufficiently supported by results or explanation.

Major concerns:

- One of the major findings is that LNMs have qualitative and quantitative immune difference to PT. However I am concerned that many of these are a consequence of sampling bias. Specifically, the collection of LN mets will almost inevitably capture uninvolved adjacent lymph node. Thus cell phenotypes that are thought to be unique to the LN metastases are likely to be an artefact of also sampling uninvolved LN. Many of the findings are consistent with this possibility, for instance that LN lymphocytes are less mature and have undergone less clonal expansion. The authors will need to use spatial methods such as mIF or spatial transcriptomics to prove that these populations are intra-tumoral.
- The authors mostly do not integrate their clusters with the literature, which is unfortunate. Instead of naming clusters by their common name, the authors use only marker genes. For instance, the CD4+CXCL13+ population that the authors spend quite some time examining are presumably T follicular Helper Cells (Tfh) which have specialised roles in lymphoid structures. However, the common name of this cluster nor its known features are not used when attempting to understand the functions of this cluster. The same applies to the stromal analysis, where the myCAF and iCAF terminology, popularised by Elyada et al and now commonly accepted, is not used. Studies such as these should make very effort to integrate data with the existing knowledge base, instead of splintering off further cell types and clusters.
- The stromal analysis is underdone, and classifies all non-endothelial cells as CAFs. Prior studies (Eg Wu et al EMBO J 2020) have shown that solid cancers including breast contain substantial populations of

cells that loosely resemble CAFs but in fact have an origin as a smooth-muscle-like cell. These cells are misclassified in the present study as CAFs.

- Similarly, the PLA2G2A+ CAFs described in this paper have been reported previously, for instance in (Wang et al Cell Research 2021) and (Chen et al Nature Comms 2020). The claims that these cells “promote immune infiltration” or “attract immune cells” are poorly supported by the evidence.

- The study is underpowered to support some of the conclusions made regarding subtype-specific differences. With only 3xHer2+ cases, the study is highly susceptible to patient-specific effects that will confound this comparison.

- The analysis of myeloid cells (Fig 5) is very descriptive and does not add substantially to the manuscript. The conclusion that myeloid populations transition between states is not well evidenced

- The final figure 7 is poorly developed, and I do not agree that pseudotime tools like Monocle can be used to infer trajectories in this setting. Analysis of aggregate datasets from different patients, different metastatic sites and disease subtypes does not seem logical. The conclusions reached, for instance that normal epithelial cells did not metastasise (7d) and that signatures of normal cells associate with prognosis (7h) are obvious.

- The analysis of migratory T cells appears to assume that T cells have migrated from the PT to LNM. This is impossible to know with the data provided, and in fact the most likely scenario is that T cells migrated to both sites from a draining lymph node, where antigen was presented to T cells by APCs. Furthermore, there is no evidence whether T cells migrate in the phenotype detected by this study or alter phenotype after migration.

Minor comments:

- Please explain the rationale and method of the STARTRAC package used for T cell analysis

- How many T cells are available for TCR analysis per patient? This will impact the ability to detect clonal expansion

- In the CAF section, it is unclear whether or when you are discussing LN vs PT

- The manuscript needs further editing for readability

- Figure text and panels are too small and the color palette is very washed out, making the data hard to visualise at times

Reviewer #3: expert in cancer biology, genomics and computational methods (Remarks to the Author):

Authors present the microenvironment analysis of the lymph node metastasis of the non-bases like breast cancer tumors using 10X and VDJ single cell RNA-sequencing by comparison of the paired primary tumor and metastasis at the lymph nodes.

Authors found that there are several distinguishing features of LN microenvironment such as T-cell characteristics (more exhausted in LN), differences in Luminal/HER2+ patients in terms of Epithelial cell dynamics, and also a novel type of CAF population.

I think the manuscript presents useful datasets and manages to summarize a lot of data into interesting results. However, there are a lot of missing details, especially for computational methods, the justification of parameters, lack of DE analysis in certain parts, unclear figures.

I list my comments below, which I hope will help improve the manuscript:

0. A lot of the main results are pushed to supplementary figures. I think some of them can be replaced with the figures in the paper.
1. It is hard to put the current manuscript in the context of many other single cell breast cancer studies. It would be great if authors could tabulate and explain the novelties and differences from other studies.
2. Although the data is made available. It would be useful to make the analysis source code available as well.
3. 40 cell types seem to be extremely subjective with markers that are selected for clustering, i.e., it would be useful to show how well these markers (for example ANXA1 in "CD4-C2-ANXA1") are expressed exclusively on these clusters.
4. Is Fig 2a the combination of PT and LN? It is not clear from the figure. Also, it is very hard to point out the 2 NK clusters (or T cell clusters for that matter) that are mentioned in the paper. It may be useful to point them out. I also do not understand the significance of Fig 2a.

5. In Fig 2a, why did authors not use all CD4 cell types? There are more CD4 cell types in Fig 1b. How were the cell types filtered?

6. Figure 2b should be updated to clearly describe which CD8 cell population corresponds to which trajectory path.

7. The PCA analysis that is performed for Fig 2c and 2d are not clear. What is the input to PCA analysis? The figure legend is not explanatory and It is not clear what is meant by "we performed a PCA to calculate the overall difference of CD8 T cells in the microenvironment of LNMTT and PT".

8. I am not sure why the authors do not present a simple differential expression analysis between PT and LN. While I can qualitatively follow the PCA analysis, it is not clear why it is necessary to follow a convoluted analysis rather than a simple DE analysis.

9. The computation of the CD8 activation scores must be clearly described.

10. While the TCGA analysis is interesting, it will be more definitive to test deconvolution of the TCGA samples using the signatures identified in the single cell datasets here using, for example, CiberSort.

11. It is not clear how much the results found in figure 2 (CD4+ T-cell signatures) and figure 3 (expansion levels) overlap with each other and stem from the same signatures providing same differences in two analyses. The details of the STARTRAC analysis must be described to ensure the features that it uses do not overlap with previous results.

12. Figure 5e legends says $\log_2(\text{FC}) > 0.5$, is this a typo? Also, at numerous places, the p-values are borderline significant, most notably PLA2G2A gene in Figure 6h.

13. There are several other typos and some of the headings are not correctly formatted. Also, Some of the sentences need to be edited because they are not clear, especially in the methods section. Below are some examples:

Line 117: "LNMTT" -> "LNMT"

Line 202: "STARAC" -> "STARTRAC"

Line 483: "UAMP embedding plot"

Line 502: "batch correlation"

Line 448: "Cells were filtered if the count of cells was lower than 300"; What does this mean?

14. The CNV analysis is very vague. The explanation in methods section needs to be clarified and justified in terms of parameter selections. I would expect the authors to apply one of the numerous CNV calling (there are at least 3) algorithms to the datasets. InferCNV is mentioned only once in the manuscript in a figure legend. Was InferCNV used for making Figure 7a,b, etc? A

15. Details of BBKNN is not explained in detail.

16. The details of diffusion map embedding technique should be described clearly with the choice and justification of the parameters.

Reviewer #4, expert in breast cancer immune microenvironment (Remarks to the Author):

The manuscript by Tong Liu et al. investigates the molecular profile of metastatic lymph nodes (LNs) and primary tumors of non-TNBC patients.

Lymph nodes of cancer patients (both non-invaded and metastatic) remain poorly studied and there is great interest in the scientific and medical community to gain knowledge on their characterization.

In this study, the authors performed scRNAseq analysis coupled with TCRseq analysis of metastatic LNs and the corresponding primary tumor of 8 treatment-naïve luminal and HER2+ breast cancer patients. The authors study approximately 118,000 unique cells, including B, CD4, CD8, NK, myeloid, epithelial, CAF, and TEC cells, classified into 40 different clusters. They analyze the proportion and the differentially expressed genes of the different cell clusters in both tissues, and perform trajectory analysis. For T cells they integrate the transcriptomic analysis to the TCR information, and evaluate clonal expansion, migration and state transition of the cells. They include in their analysis studies on cell-cell communication, some immunohistological analyses and correlation of the identified signatures with patient survival using TCGA data.

Although the manuscript represents a considerable amount of analyzed data, it appears overall as a catalog of multiple subpopulations in LNs and tumor. Many results are correlative in nature, without functional validation. Many observations confirm previously described results, and although some are new and interesting, they are drawn from too much data. The manuscript would benefit from focusing on a few of the observations and analyzing them in depth, including functional validation.

The results are sometimes approximative and over-interpreted. The manuscript suffers from lack of precision (i.e. what is the actual number of cell per cluster, per patient and per tissue?, what are the R values in fig 3 d and e; what are the units of the axis in figure 4c-e: UMIs?; how are “exhausted”, “resident memory” and the other diffusion axes defined?; the relevance of results shown in figure 3f is difficult to appreciate.

The conclusion that tumor T cells are more activated than LN T cells because there are less naïve cells in the tumor, does not consider that naïve T cells are normally present in LNs (even in healthy individuals).

The text should be reviewed by a native English speaker, as it contains many mistakes.

REVIEWER COMMENTS

Reviewer #2, expert in breast cancer microenvironment and single cell RNA-seq (Remarks to the Author):

This manuscript reports on the analysis by scRNA-Seq of 8 primary breast cancers (PT) and paired metastatic lymph nodes (LNM). Overall the study is well collected and the data is of good quality, but the analysis is unfocussed and incomplete. Lymph node metastasis (LNM) is a common occurrence in breast cancer, but little data exists comparing the cells found in LNM and PT. This will therefore be a useful resource for those in the field.

The authors take a broad 'landscape' approach to describe the immune, stromal and epithelial components of these datasets. Much of the manuscript focusses on immune cells, using both scRNA-Seq and TCR sequencing data. The clustering is well performed and consistent with other literature. However, the manuscript fails to reach conclusive findings a number of times, and claims are often not sufficiently supported by results or explanation.

Major concerns:

1. One of the major findings is that LNMs have qualitative and quantitative immune difference to PT. However I am concerned that many of these are a consequence of sampling bias. Specifically, the collection of LN mets will almost inevitably capture uninvolved adjacent lymph node. Thus cell phenotypes that are thought to be unique to the LN metastases are likely to be an artefact of also sampling uninvolved LN. Many of the findings are consistent with this possibility, for instance that LN lymphocytes are less mature and have undergone less clonal expansion. The authors will need to use spatial methods such as mIF or spatial transcriptomics to prove that these populations are intra-tumoral.

Response:

Thanks for the important comment and suggestion. Firstly, the appearance of normal lymph nodes is different from the lymph nodes with tumor infiltration. The degree of tumor cell infiltration of surgery samples from patients was discreetly evaluated and then sequenced. Moreover, scRNA-Seq was also used to further confirm the tumor cell infiltration of LNMs.

At the suggestion of Reviewer, we randomly picked LN tissues of 4 patients from the 8 patients in the present study to perform spatial transcriptomics to demonstrate the cell populations we analyzed are intra-tumoral. To integrate the data, scRNA-seq transcriptome and spatial transcriptome have been preprocessed by SCTransform function and PCA analysis. Then FindTransferAnchors and TransferData functions were used to measure each cluster score for each spot. Description of methods has been updated in page 22 paragraph 3.

The data of the spatial transcriptomics showed that the ratio of tumor cells is over 50% in all the 4 patients and over 75% in 3 of them, suggesting that the samples are from LN metastases (please refer to the following Fig. a & b, which are shown in the new version of Fig. 1f). Corresponding descriptions in the Results part have been added on page 5, paragraph 1.

Furthermore, we analyzed the distribution of immune cells in LNMT and found that a large amount of immune cells are infiltrated around tumor cells, which are mainly plasma B, CD4 T and CD8 T cells. However, we found that most of T cells in LNMT are Naïve or in early developmental state (please refer to the following Fig c). We then mapped our single cell sequencing signature into spatial transcriptome and confirmed that these naïve T cells have low activity. In conclusion, the data of spatial transcriptomics proved that the samples thrown to single cell sequencing are mainly from lymph node metastasis and further demonstrated that T cells around tumors are less active (Fig c).

a. Distribution of epithelial cells in LNMT by Spatial transcriptome integrated with scRNA-Seq

b. Percentage of spots representing tumor cells detected in LNMT.

c. Spatial transcriptome analysis showed the distribution of epithelia cell, naïve CD4 T cells (CD4-C1-RPL) and naïve CD8 T cells (CD8-C1-CD8D) in LNMT.

2. The authors mostly do not integrate their clusters with the literature, which is unfortunate. Instead of naming clusters by their common name, the authors use only marker genes. For instance, the CD4+CXCL13+ population that the authors spend quite some time examining are presumably T follicular Helper Cells (Tfh) which have specialised roles in lymphoid structures. However, the common name of this cluster nor its known features are not used when attempting to understand the functions of this cluster. The same applies to the stromal analysis, where the myCAF and iCAF terminology, popularised by Elyada et al and now commonly accepted, is not used. Studies such as these should make every effort to integrate data with the existing knowledge base, instead of splintering off further cell types and clusters.

Response:

Thanks for the suggestion. With the prevalence of the single cell technology and increase in resolution, more and more cell types and clusters are discovered, defined and reported. However, we found that the existing definition or clusters cannot fully satisfy our research objective. To better understand the heterogeneity of tumor microenvironment, we defined some clusters according to the specific genes distinguished them.

As suggested, we compared our clusters names with the literature and integrated our data with the existing knowledge base, which are annotated on the graph including CD4, CD8 and CAFs (please refer to the following images which are shown in the new version of Fig. 2a, Supplementary Fig. 6a, and Fig. 6a). Corresponding descriptions in the Results part have been added on page 6, paragraph 2 and on page 11, paragraph 2.

Fig 2a

Fig 6a

3. The stromal analysis is underdone, and classifies all non-endothelial cells as CAFs. Prior studies (Eg Wu, et al., EMBO J, 2020) have shown that solid cancers including breast contain substantial populations of cells that loosely resemble CAFs but in fact have an origin as a smooth-muscle-like cell. These cells are misclassified in the present study as CAFs.

Response:

Thanks for the important advice. According to Wu et al’s report, three types of stromal cells were identified in TNBC including endothelial cells, CAFs and perivascular-like cells (PVLs) (Eg Wu et al, EMBO J 2020). These type of PVLs share many identical genes with CAFs such as ACTA2, PDGFRB and matrix producing genes. We are sorry for misclassifying perivascular-like cells (CAF-C4-ACTA2, CAF-C5-MYH11) into CAFs in our study. We have corrected them in the new version of Fig. 1b, Fig. 1e, Fig. s1c, Fig. s1d, Fig. s1e, Fig. s1f, Fig. 6a, Fig. 6c, which do not affect our conclusion. Corresponding descriptions in the Results part have been added on page 5, paragraph 1 and page 11, paragraph 2. Thanks again for pointing out this misclassification.

4. Similarly, the PLA2G2A+ CAFs described in this paper have been reported previously, for instance in (Wang et al Cell Research 2021) and (Chen et al Nature Comms 2020). The claims that these cells “promote immune infiltration” or “attract immune cells” are poorly supported by the evidence.

Response:

Thanks a lot for the important advice. In Chen's study, PLA2G2A+ CAFs belong to iCAF which may promote tumor cell survival and proliferation. iCAFs were heterogeneous and classified into four sub-clusters (Chen et al Nature Comms 2020). In our dataset, iCAF can be divided into two types including PLA2G2A+ CAFs and APOD+ CAFs. We supplied several evidences to support that PLA2G2A+ CAFs may promote immune infiltration: 1)

PLA2G2A+ CAFs were characterized by high expression of OGN, which encodes a small leucine-rich proteoglycan protein functioning in T-cell recruitment and immune infiltration in tumors (Hu X et al EBioMedicine 2018); 2) The cell-cell interaction analysis showed that PLA2G2A+ CAFs displayed strong interaction with immune cells such as T cells and macrophages (Fig. 6d); 3) The data of the receptor-ligand interaction analysis revealed that highly expressed genes such as PLA2G2A, OGN, VSIR, DPP4 in PLA2G2A+ CAFs interacts with the receptors of immune cells HLA-DRB1, $\alpha 4\beta 1$ integrin, CCL4L2 and CCL3L1 respectively (Fig. 6d, 6e); 4) TCGA database from 640 patients (LumA, LumB, Her2+) showed that PLA2G2A was positively correlated with the CD45 antigen PTPRC, B cell marker CD79A and CD8 T cells marker CD8A in breast cancer patients (Fig. 6g, Fig. s6d); 5) Immunofluorescence analysis showed that PLA2G2A+ CAFs had a similar spatial distribution with macrophages and CD8 T cells (Fig. 6j), and confirmed the interactions of PLA2G2A+ CAFs with macrophages and CD8 T cells at the single-cell level (Fig. 6k).

However, there is indeed lack of experiments to improve our conclusion. To further confirm the relationship between PLA2G2A and immune infiltration, we supplemented experiments in vitro. Monocytes THP1 were treated with PLA2G2A protein for 4 hours and then cell migration was detected by Transwell. We found that PLA2G2A significantly promoted cell migration of monocytes, which is shown in the new version of Fig. 6f. Corresponding descriptions in the Results part have been added on page 12, paragraph 1. We hope the supplementary experiment can at least partially answer this Reviewer's question.

PLA2G2A promotes the migration of THP1 cells

Transwell assays were used to measure cell migration. PLA2G2A (0.5 $\mu g/ml$) was respectively applied to the top chamber and the bottom chamber, then cells were incubated at 37°C in 5% CO₂ for 4h. Statistical testing was performed by T-test. (***: $P < 0.001$, ****: $P < 0.0001$)

5. The study is underpowered to support some of the conclusions made regarding subtype-specific differences. With only 3xHer2+ cases, the study is highly susceptible to patient-specific effects that will confound this comparison.

Response:

The single cell sequencing analysis of aggregate datasets found that PLA2G2A+ CAFs have a high frequency in HER2+ tumors (Fig. 6b& c). To avoid that this finding is caused by enrichment in one patient, we also examined the ratio of PLA2G2A+ CAFs in each patient. We found that PLA2G2A+ CAFs are exclusively enriched in all the three HER2+ cases (please refer to the following Figure), suggesting the finding may not be patient-specific effects driven by single patient. However, we agree with the reviewer that the data from 3 patient is not enough to support our conclusions about histological characteristics of Her2+ cases. In addition to the single cell sequencing, we supplemented the database analysis and immunohistochemistry to verify the relationship between PLA2G2A and HER2. Analysis of the TCGA database showed that PLA2G2A was highly expressed in 67 HER2+ patients. We then validated this finding in 12 HER2+ and 11 luminal breast cancer patients by IHC staining. The results also proved that PLA2G2A+ CAFs were more abundant in HER2+ tumors than in luminal tumors (Fig. 6g & h of previous version). In this revised edition, we further supplemented 9 HER2 and 9 luminal cases to detect PLA2G2A expression. And the results similarly demonstrated that PLA2G2A was highly expressed in HER2+ patients. We integrated the two experiments and the data are shown in the new version of Fig. 6i. We hope that combining the results of single cell sequencing with validation experiments can adequately support our conclusion.

UMAP embedding plot showed the distribution of CAFs in each HER2+ patient. The cells circled are PLA2G2A+ CAFs.

6. The analysis of myeloid cells (Fig 5) is very descriptive and does not add substantially to the manuscript. The conclusion that myeloid populations transition between states is not well evidenced

Response:

We agree with this reviewer's opinion. There is lack of solid evidence to support myeloid populations transition between states in this study. And the analysis of myeloid populations transition indeed does not add substantially to the manuscript. Therefore, we deleted this part of results in the revised version. However, we supplemented the analysis to compare macrophages between LNMT and paired PT. It is well known that M1 and M2 signatures represent the different functions of macrophages in tumor. We found there are no difference in the ratio of these 2 types of macrophages in PTs compared to LNMTs (please refer to the following Fig a, which is shown in the new version of Fig. S5g). However, using differential gene analysis, we found that macrophages have a higher expression of IFI27, IFITM1, and IFI44L in PTs than in LNMTs, suggesting activation of INF- γ signaling in PTs. Furthermore, CCL8, which has been reported to promote breast metastasis and progression (Kalluri R Nat Rev Cancer 2016), was more highly expressed in PTs than in LNMTs, indicating distinct microenvironments in PT and LNMT that could influence tumor cell survival (please refer to the following Fig b, which is shown in the new version of Fig 5j). Corresponding descriptions in the Results part have been added on page 11, paragraph 1.

a. Correlation between M1 signature and M2 signature of macrophages in PT and LNM

b. Differential gene analysis of M1 signature and M2 signature of macrophages in PT and LNM

7. The final figure 7 is poorly developed, and I do not agree that pseudotime tools like Monocle can be used to infer trajectories in this setting. Analysis of aggregate datasets from different patients, different metastatic sites and disease subtypes does not seem logical. The conclusions reached, for instance that normal epithelial cells did not metastasize (7d) and that signatures of normal cells associate with prognosis (7h) are obvious.

Response:

1) As several studies have shown that epithelial tumor cells may originate from normal epithelial cells in breast cancer (Polyak J Clin Invest 2007), we assumed that tumor cells may share partial transcriptome program with normal or normal-like epithelial cells. We agree with the reviewer that it is challenged to infer trajectories from different samples by using pseudotime tools because of the patient-specific program in tumor cells. Therefore, we integrated the expression matrix of epithelial cells by BBKNN, which may neutralize patient specific effects including different patients and disease subtypes. Then cell clusters were determined by Leiden function with resolution = 0.8 in batched-removed epithelial cells. We compared each cell cluster with other clusters by FindAllMarkers function in Seurat package to determine the batch-removed differential expressed genes, and top 20 differential expressed genes per cluster were used to order cells in Monocle (please refer to the following Fig a left, which is shown in Fig 7c). The detail of this method has been updated in page 23 paragraph 3. To further confirm the results, we utilized another way to remove batch effect in Monocle analysis (Kim N et al Nat Commun 2020). We ordered epithelial cells based on gene list, which includes signatures of normal epithelial cells in different state in normal breast (Nguyen et al Nat Commun 2018) (please refer to the following Fig a, right). The above two types of analysis showed a similar developmental path of epithelial cells in Monocle analysis. Furthermore, we showed the DDRtree plot for each patient and found that none of the patients dominated in one specific state, which means that there is no large batch effect in this analysis. (please refer to the following Fig b, which is shown in the new version of Fig S7d).

a. Monocle analysis showed that distribution of epithelial cells in different setting of parameters.

b. Monocle analysis showed the distribution of epithelial cells on DDRtree for each patient

2) Our purpose of performing this analysis is to study the transcriptome changes as the normal-like epithelial cells development towards different malignant epithelial cells state. We have deleted the inappropriate conclusion “normal epithelial cells did not metastasize”. Fig 7h in the previous version was removed and the conclusion that “High levels of population 3 signature genes predicted a better prognosis in TCGA-BRCA” was deleted.

8. The analysis of migratory T cells appears to assume that T cells have migrated from the PT to LNM. This is impossible to know with the data provided, and in fact the most likely scenario is that T cells migrated to both sites from a draining lymph node, where antigen was presented to T cells by APCs. Furthermore, there is no evidence whether T cells migrate in the phenotype detected by this study or alter phenotype after migration.

Response:

As the same clones of T cells appear in PT and LNM tissues, we named these cells as “migrated T”. We agree with this reviewer that "migrated T" may be transferred from other drain LNs or other tissues. However, regardless of the source of "migrated T", we showed that T cells transferred to lymph node metastasis microenvironment are more immunosuppressed, which further supported our conclusion by utilization of TCR analysis. We revised the inappropriate descriptions shown in page 9 paragraph 1. Additionally, to avoid the assumption that T cells have migrated from the PT to LNM, we renamed these cells as "Matched T" according to another study (Pauken et al J Exp Med 2021).

Minor comments:

1. Please explain the rationale and method of the STARTRAC package used for T cell analysis

Response:

STARTRAC was used to analyze different properties and functions of T cells based on paired single cell transcriptomes and TCR sequences (Zhang et al Nature 2018). The inputs of STARTRAC are TCR clonal information and cell cluster information of each T cell. The most important role of STARTRAC is to quantify the expansion, transition and migration ability of T cells, which cannot be inferred by traditional gene signature analysis. In our dataset we used STARTRAC-expa and STARTRAC-tran to quantify the expansion and transition of T cells.

STARTRAC-expansion is usually used in the standard TCR clonality measurement but was specifically applied to different T cell clusters in our analyses. Normalized Shannon entropy was used to calculate the evenness of the TCR repertoire of the given T cell cluster and then define the STARTRAC-expansion index as 1-evenness. Mathematically, the STARTRAC-expa index of a specific cluster with N clonotypes is defined by the following formula:

$$I_{expa}^{STARTRAC} = 1 - \frac{-\sum_1^N p_i \log_2 p_i}{\log_2 N}$$

p_i represents the cell frequency of clonotype in the cluster, and a clonotype is defined by identical, full-length, paired α and β TCR chains. STARTRAC-expansion ranges from 0 to 1, with 0 representing no clonal expansion for each clonotype and 1 representing a cluster composed of only 1 clonally expanded clonotype. A high STARTRAC-expansion indicated high clonality.

STARTRAC-transition was used to quantify the extent of state transition of each clonotype within a given cluster. The STARTRAC-transition index at the cluster level was defined as the weighted average of all TCR clonotype state transition indices contained in the cluster.

$$I_{tran}^{STARTRAC} = \sum_{t=1}^T p_{cls}^t I_{tran}^t$$

p_{cls}^t is the ratio of the number of cells with clonotype t in cluster cls to the total number of cells in cluster cls.

Methods have been updated in page 25.

2. How many T cells are available for TCR analysis per patient? This will impact the ability to detect clonal expansion.

Response:

59,327 immune T cells were captured and 47803 of them were performed TCR analysis. The following figure showed the number of T cells for TCR analysis in each patient, in which very rare immune cells were found in Patient 2 and Patient 4, thus these two samples were removed from downstream analysis. The figure has been updated in Fig. S3a. Corresponding

description has been updated in page 8 paragraph 1.

Bar plot showed the number of T cells for TCR analysis in each patient, in which very rare immune cells were found in patient 2 and patient 4.

3. In the CAF section, it is unclear whether or when you are discussing LN vs PT

Response:

In the previous version, the ratio of CAF subclusters in LN vs PT of all patients was shown in Fig 1e. We supplemented the comparison of the proportion of total CAFs in PT and LNMT. We found there is no difference (please refer to the following Fig a, which is shown in the new version of Fig 1d). We further particularly compared the percentage of each CAFs subcluster within stromal cells between PT and LNM in each patient. The results also showed there is no difference (please refer to the following Fig b). Corresponded description has been updated in page 11 paragraph 2.

a. Percentage of CAFs clusters between PT and LNM. Statistical analysis was performed by two-sided Wilcoxon test (n= 8).

b. Percentage of CAFs subclusters within stromal cells between PT and LNM. Statistical analysis was performed by two-sided Wilcoxon test (n= 8).

4. The manuscript needs further editing for readability

Response:

As suggested, the manuscript has been read and edited by a native English speaker, we hope it has been improved.

5. Figure text and panels are too small and the color palette is very washed out, making the data hard to visualise at times

Response:

Thanks for the advice! we have adjusted the text size and the contrast of our figures, which are more readable now.

Reviewer #3: expert in cancer biology, genomics and computational methods (Remarks to the Author):

Authors present the microenvironment analysis of the lymph node metastasis of the non-bases like breast cancer tumors using 10X and VDJ single cell RNA-sequencing by comparison of the paired primary tumor and metastasis at the lymph nodes.

Authors found that there are several distinguishing features of LN microenvironment such as T-cell characteristics (more exhausted in LN), differences in Luminal/HER2+ patients in terms of Epithelial cell dynamics, and also a novel type of CAF population.

I think the manuscript presents useful datasets and manages to summarize a lot of data into interesting results. However, there are a lot of missing details, especially for computational methods, the justification of parameters, lack of DE analysis in certain parts, unclear figures.

I list my comments below, which I hope will help improve the manuscript:

0. A lot of the main results are pushed to supplementary figures. I think some of them can be replaced with the figures in the paper.

Response:

Because of the abundant data and limited space, we have to put many figures in the supplementary figures. However, as suggested, we have adjusted two of them to the figures in the paper. Fig s6e is adjusted to Fig 6k and Fig s7d is adjusted to Fig 7h.

1. It is hard to put the current manuscript in the context of many other single cell breast cancer studies. It would be great if authors could tabulate and explain the novelties and differences from other studies.

Response:

Thanks for the good suggestion. We tabulate the main findings of other studies on single-cell RNA-seq in breast cancer and explain the novelties and differences of our study.

Title of reported Single-cell studies in breast cancer	Major conclusion
Single-cell RNA-seq enables comprehensive tumour and immune cell profiling in primary breast cancer	This study collected 11 patients with a total of 515 cells, and the use of single cell sequencing reveals the heterogeneity of the microenvironment in primary breast cancer.
Spatially and functionally distinct subclasses of breast cancer-associated fibroblasts revealed by single cell RNA sequencing	The heterogeneity and function of several CAFs in tumors were identified in mouse models of breast cancer.
Chemoresistance Evolution in Triple-Negative Breast Cancer Delineated by Single-Cell Sequencing	The single cell DNA sequencing was used to reveal the cloned heterogeneity of tumor in different chemotherapy phases, drug resistance of tumor clones in the treatment pressure.

Single-cell profiling of breast cancer T cells reveals a tissue-resident memory subset associated with improved prognosis	scRNA-seq technology analyzed T cells in tumor microenvironment of primary breast cancer and found that a group of tissue resident T cells were related to prognosis.
Defining the emergence of myeloid-derived suppressor cells in breast cancer using single-cell transcriptomics	A mouse model of breast cancer was used to reveal the characteristics of myeloid cells infiltrated in breast cancer.
Transcriptional diversity and bioenergetic shift in human breast cancer metastasis revealed by single-cell RNA sequencing	PDX mouse model was used to reveal transcriptional characteristics after metastasis of TNBC. It was found that the oxidative phosphorylation pathway was activated in TNBC after metastasis
The novelties of our study	For the first time, the tumor microenvironment of primary tumor and paired lymph node metastasis tissues were analyzed systematically and compared by scRNA-Seq in 8 breast cancer patient tissues. We found that the lymph node microenvironment had stronger immunosuppression compared with the primary tumor, and revealed the potential mechanism. We also identified the function of PLA2G2A+ CAFs in promoting immune infiltration in the breast microenvironment.

2. Although the data is made available. It would be useful to make the analysis source code available as well.

Response:

We have consolidated the key code which has been uploaded on Github. (<https://github.com/bio-liucheng/brca-singlecell>).

3. 40 cell types seem to be extremely subjective with markers that are selected for clustering, i.e., it would be useful to show how well these markers (for example ANXA1 in "CD4-C2-ANXA1") are expressed exclusively on these clusters.

Response:

Thanks for the good suggestion.

1) We think "cell clusters" is more suitable than "cell types", because annotation of cell cluster does not intend to identify novel cell types but to describe the different cell state in our dataset and investigated the difference of cell clusters across group comparison. We have adjusted the wording.

2) 40 cell clusters are obtained through unified parameters. For T cells, NK cells, stromal cells and myeloid cells, the cell clusters were annotated by parameter with a resolution of 0.8 in scanpy packages. It is not a subjective classification.

3) Clusters were annotated based on the expression of known marker genes or the most highly expressed genes. As suggested, in order to display the selected marker genes expression level across different clusters, the heatmap of markers genes in 40 clusters has been shown below. The relative expression of marker genes or highly expressed genes for each cluster has been supplied in supplement table 2.

a. Heatmap showed the marker genes or highly expressed genes across cell clusters.

4. Is Fig 2a the combination of PT and LN? It is not clear from the figure. Also, it is very hard to point out the 2 NK clusters (or T cell clusters for that matter) that are mentioned in the paper. It may be useful to point them out. I also do not understand the significance of Fig 2a.

Response:

We are sorry for the misleading Figure. Fig. 2a showed lymphocytes cells excluding B cells combination of PT and LN. However, at the reminding of the reviewer, we think it will be clearer and more useful to show the distribution of immune cells in PT and LN separately. Thus, Fig. 2a is replaced with a new one. Furthermore, as suggested, we labeled the numbers of each cluster on a UMAP embedding plot to distinguish these cell types well.

5. In Fig 2a, why did authors not use all CD4 cell types? There are more CD4 cell types in Fig 1b. How were the cell types filtered?

Response:

In all our dataset, we divided CD4 T cells into seven types, which are shown in Fig 1b and Fig 2a. There are no differences in CD4 T cells types between Fig 1b and Fig 2a. To better show the distribution of CD4 T clusters, cells clusters in UMAP embedding were numbered in the new version of Fig 2a.

6. Figure 2b should be updated to clearly describe which CD8 cell population corresponds to which trajectory path.

Response:

As suggested, Figure 2b has been updated to clearly show CD8 cell population and the trajectory path. As shown in the figure, CD8-C1-CD8B (12) was the initial stage of CD8+ T differentiation, then it differential to CD8-C2-CCL5 (13), which further branched into CD8-C3-GZMK (14) or CD8-C4-HSPA1A (15). At last, CD8-C3-GZMK differentiated to CD8-C5-CXCL13 (16).

7. The PCA analysis that is performed for Fig 2c and 2d are not clear. What is the input to PCA analysis? The figure legend is not explanatory and It is not clear what is meant by "we performed a PCA to calculate the overall difference of CD8 T cells in the microenvironment of LNMTT and PT".

Response:

We are sorry for the unclear figures. In Fig 2c and 2d, the expression matrix of CD8 T cells from PT and LNM are input to PCA analysis. To compare the difference of PT and LNMT in each PC, Z scores from unpaired-wise t-test were measured between tumor and lymph node for each PC and showed the first 20 PCs in Fig 2c. Fig 2d showed the distribution of CD8 T cells in PC1 and PC2 spaces from PCA analysis.

Fig 2c has been moved to supplementary Fig. 2f in the revised version. Description of method has been updated in the figure legend of supplementary Fig. 2f.

8. I am not sure why the authors do not present a simple differential expression analysis between PT and LN. While I can qualitatively follow the PCA analysis, it is not clear why it is necessary to follow a convoluted analysis rather than a simple DE analysis.

Response:

The rationality of PCA analysis in CD8 T analysis is that we convoluted the variance into PCs and figured out the most significant PC to distinguish CD8 T between PT and LNMT, which qualitatively guides us the direction for downstream analysis. PCA analysis can project the potential biological differences to PCs, which may obtain independent and interpretable biological effects in each PC. Differential expression analysis is a comprehensive difference result, which may come from the mixing of multiple pathways or subpopulations in various cell states. Therefore, we prefer to perform direct differential analysis of cell types within similar states, which can better reveal the biological effects with specific cell types. In the comparison of widely distributed cell states, PCA analysis is able to deconvolute the difference to the PCs and make it interpretable. For example, in our analysis, we found that the PC2 can most separate CD8 T cells between PT and LNMT, indicating that PC2 may be the most important component of distinguishing CD8 T cells of PT from LNMT.

9. The computation of the CD8 activation scores must be clearly described.

Response:

In detail, we used the average expression levels of marker genes to represent the CD8 activation score. The CD8 T activation marker genes are CD69, CCR7, CD27, BTLA, CD40LG, IL2RA, CD3E, CD47, EOMES, GNLY, GZMA, GZMB, PRF1, IFNG, CD8A, CD8B, FASLG, LAMP1, LAG3, CTLA4, HLA-DRA, TNFRSF4, ICOS, TNFRSF9, TNFRSF18, which are referenced from a published study (Steven Breast Care (Basel) 2018). The list of all signature genes has been added in supplement table 3.

10. While the TCGA analysis is interesting, it will be more definitive to test deconvolution of the TCGA samples using the signatures identified in the single cell datasets here using, for example, CiberSort.

Response:

Thanks for the importance suggestion. Following procedures recommended by the developers, CiberSortx was applied in Her2 and luminal subtypes samples from TCGA

BRCA cohort, to score cell type fraction. Signature matrix was created online (<https://cibersortx.stanford.edu/>) based on our gene expression matrix by default parameters. We divided the TCGA cohort into two groups with the highest 20% and the lowest 20% of PLA2G2A expression to estimate the relationship between cell type fractions and PLA2G2A expression levels. And we found that samples with higher PLA2G2A expression has higher frequency of CD8+ T cell, Macrophages, DCs, plasma B and TECs.

a. The difference of frequency of representative cell types in TCGA samples between PLA2G2A high and low groups. Statistical testing was performed by T-test. (ns no significant, * $P < 0.05$, ** $P < 0.01$, ***: $P < 0.001$, ****: $P < 0.0001$)

11. It is not clear how much the results found in figure 2 (CD4+ T-cell signatures) and figure 3 (expansion levels) overlap with each other and stem from the same signatures providing same differences in two analyses. The details of the STARTRAC analysis must be described to ensure the features that it uses do not overlap with previous results.

Response:

1) As for the analysis of CD4+ T-cell signatures in figure 2, the main findings are the distinct differentiation path and functions of CD4+CXCL13+ cells in two microenvironments. We found that CD4+CXCL13+ T cells are more likely reprogrammed to an exhausted state in LNMT. In figure 3, we used STARTRAC-expansion to measure the expansion of each T cell cluster. The results showed that CD4-C6-CXCL13 had the highest expansion ability among CD4 T cells and displayed higher expansion ability in PT compared to LNMT. We do not think there is any overlap between Fig 2 and Fig 3, because we used different analyses and obtained the results from different perspectives. However, the conclusion of Fig 2 and Fig 3 is similar, we demonstrated that CD4+ T cells in LNMT are less mature and active than those in PT.

2) STRATRAC was used to analyze different properties and functions of T cells based on paired single cell transcriptomes and TCR sequences. The input of STARTRAC are TCR clonal information and cell types information of each T cell. The most important role of STARTRAC is to quantify the expansion, transition and migration ability of T cells, which cannot be inferred by traditional gene signature analysis. Methods of STARTRAC have been updated in page 25 paragraph 2.

12. Figure 5e legends says $\log_2(\text{FC}) > 0.5$, is this a typo? Also, at numerous places, the p-values are borderline significant, most notably PLA2G2A gene in Figure 6h.

Response:

The significant threshold in differential analysis in this study was set as p-value < 0.05 and $\log_2(\text{fold change}) \geq 0.5$. Because 10x sequencing technology has low mRNA capture rate, it is common to set threshold like this (Rodda et al Immunity 2018).

The borderline significance may be resulted from limited cases. As for Figure 6h, 12 luminal tumor tissues and 11 HER2 tumor tissues were stained with PLA2G2A and scored. To reduce the impact of limited cases, we additionally added 9 cases respectively in Luminal and HER2+ patients to detect the expression of PLA2G2A protein by Immunohistochemistry (IHC). We integrated the two experiments and the data are shown in the new version of Fig. 6i. (Fig a).

a. PLA2G2A score of IHC in luminal or HER2+ breast cancer patients. Two-sided Wilcoxon test.

13. There are several other typos and some of the headings are not correctly formatted. Also, Some of the sentences need to be edited because they are not clear, especially in the methods section. Below are some examples:

Line 117: "LNMTT" -> "LNMT"

Line 202: "STARAC" -> "STARTRAC"

Line 483: "UAMP embedding plot"

Line 502: "batch correlation"

Line 448: "Cells were filtered if the UMI count of cell was lower than 300"; What does this mean?

Response:

We are sorry for the errors. We have corrected them in the revised edition and also carefully checked the other descriptions in the Methods part. As for "Cells were filtered if the UMI count of cell was lower than 300", we wanted to filter low quality cells which have low mRNA capture rate or dying cells. "Cells were filtered if the UMI count of cell was lower than 300" has been corrected into "Cells were filtered if their UMI counts were lower than 300" in page 19 paragraph 2.

14. The CNV analysis is very vague. The explanation in methods section needs to be clarified and justified in terms of parameter selections. I would expect the authors to apply one of the numerous CNV calling (there are at least 3) algorithms to the datasets. InferCNV is mentioned only once in the manuscript in a figure legend. Was InferCNV used for making Figure 7a,b, etc?

Response:

Thanks for the importance suggestion. As suggested, we used another CNV calling package, CopyKAT (Gao et al., 2021) to estimate the aneuploid of epithelial cells, which were considered as malignant cells. Gene counts matrix of epithelial cells was extracted from the Seurat object. Quality control filtering was performed with at least one gene in each chromosome and then to calculate DNA copy numbers. At least 100 genes per segment were applied to keep up with the inferCNV parameters. KS.cut was selected as 0.05 to increase the sensitivity.

We found that CopyKAT and inferCNV can classify the same epithelial cells into malignant cells in our study. CopyKAT may set a higher standard which may filter out more malignant cells. As our aim is to investigate the developmental path of epithelial cells in tumor, it will not impact our conclusion in Fig 7.

Yes, inferCNV is also used for making Fig. 7a&b. We have added the description in the figure legend.

a. DDRtree plot showed epithelial cells classified as malignant cells by CopyKAT (left) or inferCNV (right)

15. Details of BBKNN is not explained in detail.

Response:

Primary tumor and paired lymph node samples were collected, library constructed and sequencing synchronously for each patient. Theoretically, batch effect between primary tumor and paired lymph node are less than that between different patients. Because BBKNN's main assumption is that at least some cells of the same type exist across batches, and that the differences between the same cell type across batches caused by batch effects are less than the differences between cells of different types within a batch (Polanski et al Bioinformatics 2020). Thus, we assume that BBKNN integration method is more suitable for our datasets as different patients may share most cell types. BBKNN modifies the neighborhood construction step to produce a graph that is balanced across all batches of the data. This approach treats the neighbor network as the primary representation of the data. For each cell, the BBKNN graph is constructed by finding the k-nearest neighbors for each cell in each user-defined batch independently. As for our data, we implemented BBKNN from the scanpy package by using the "bbknn" function", and the parameters were set as follows: batch_key = "patient", n_pcs= 35.

Method has been updated in page 20 paragraph 1.

16. The details of diffusion map embedding technique should be described clearly with the choice and justification of the parameters.

Response:

Diffusion map embedding was used for speculating cell differentiation path and the connection between cell types. Compared with UMAP embedding, diffusion map embedding is more focused on revealing local relationship and cluster structure. Thus, we only implemented diffusion map on cell types that have the same lineage, such as CD8 T cells. Diffusion map function was implemented in scanpy packages. The BBKNN batch corrected expression matrix was used for calculating neighbor matrix by following parameters: n_neighbors = 10, n_pcs = 15 and method = "gauss". Then the neighbor matrix was used to

calculate diffusion map. We found that changing the number of neighbours dose not impact the relative position of cell types in diffusion map. Methods has been updated in page 22 paragraph 3.

Reviewer #4, expert in breast cancer immune microenvironment (Remarks to the Author):

The manuscript by Tong Liu et al. investigates the molecular profile of metastatic lymph nodes (LNs) and primary tumors of non-TNBC patients.

Lymph nodes of cancer patients (both non-invaded and metastatic) remain poorly studied and there is great interest in the scientific and medical community to gain knowledge on their characterization.

In this study, the authors performed scRNAseq analysis coupled with TCRseq analysis of metastatic LNs and the corresponding primary tumor of 8 treatment-naïve luminal and HER2+ breast cancer patients. The authors study approximately 118,000 unique cells, including B, CD4, CD8, NK, myeloid, epithelial, CAF, and TEC cells, classified into 40 different clusters. They analyze the proportion and the differentially expressed genes of the different cell clusters in both tissues, and perform trajectory analysis. For T cells they integrate the transcriptomic analysis to the TCR information, and evaluate clonal expansion, migration and state transition of the cells. They include in their analysis studies on cell-cell communication, some immunohistological analyses and correlation of the identified signatures with patient survival using TCGA data.

Although the manuscript represents a considerable amount of analyzed data, it appears overall as a catalog of multiple subpopulations in LNs and tumor. Many results are correlative in nature, without functional validation. Many observations confirm previously described results, and although some are new and interesting, they are drawn from too much data. The manuscript would benefit from focusing on a few of the observations and analyzing them in depth, including functional validation.

The results are sometimes approximative and over-interpreted. The manuscript suffers from lack of precision (i.e. what is the actual number of cell per cluster, per patient and per tissue?, what are the R values in fig 3 d and e; what are the units of the axis in figure 4c-e: UMIs?; how are “exhausted”, “resident memory” and the other diffusion axes defined?; the relevance of results shown in figure 3f is difficult to appreciate.

The conclusion that tumor T cells are more activated than LN T cells because there are less naïve cells in the tumor, does not consider that naïve T cells are normally present in LNs (even in healthy individuals).

Response:

Thanks very much for the comments, encouragement and important suggestions. We have answered the questions one by one through the following specific explanations and solutions.

- 1) The manuscript would benefit from focusing on a few of the observations and analyzing them in depth, including functional validation.

Response:

It is a very good suggestion. we established the link between PLA2G2A+ CAFs and immune cells through single cell sequencing analysis, TCGA-BRCA dataset and Immunofluorescence. To further validate that PLA2G2A+ CAFs promote immune infiltration, we supplemented experiments in vitro. Monocytes THP1 were treated with PLA2G2A protein for 4 hours and

then cell migration was detected by Transwell. We found that PLA2G2A significantly promoted cell migration of monocytes, which are shown in the new version of Fig. 6f. Corresponding descriptions in the Results part have been added on page 12, paragraph 1. We will perform more experiments to investigate the role of PLA2G2A in immune infiltration.

PLA2G2A promotes migration of THP1 cells

Transwell assays were used to measure cell migration. PLA2G2A (0.5 μg/ml) was respectively applied to the top chamber and the bottom chamber, then cells were incubated at 37°C in 5% CO₂ for 4h. Statistical testing was performed by T-test. (***: P < 0.001, ****: P < 0.0001)

2) The results are sometimes approximative and over-interpreted.

Response:

We carefully checked the results and found some conclusions we reached are not well evidenced. We adjusted some of them. (1) In the previous version of manuscript, migratory T cells were considered as T cells migrated from the PT to LNM. However, "migrated T" may be transferred from other drain LNs or other tissues. Furthermore, there is no evidence whether T cells migrate in the phenotype detected by this study or alter phenotype after migration. We revised the inappropriate descriptions shown in page 9 paragraph 1. However, the adjustment does not affect the conclusion of this study. Regardless of the source of "migrated T", we showed that T cells transferred to lymph node metastasis microenvironment are more immunosuppressed. Additionally, to avoid the assumption that T cells have migrated from the PT to LNM, we renamed these cells "Matched T" according to another study (Pauken et al J Exp Med 2021).

(2) We think some results in figure 7 are not well evidenced. It is challenged to infer trajectories from different samples by using pseudotime tools because of the patient-specific program in tumor cells. Therefore, we integrated the expression matrix of epithelial cells by BBKNN, which may neutralize patient specific effects including different patients and disease subtypes. We compared each cell cluster with other clusters to determine the batch-removed differential expressed genes, and all differential expressed genes were used to order cells in Monocle (please refer to the following Fig a left, which is shown in Fig 7c). To further confirm the results, we utilized another way to remove batch effect in Monocle

analysis. We ordered epithelial cells based on gene list, which includes signatures of normal epithelial cells in different state in normal breast (Nguyen et al Nat Commun 2018) (please refer to the following Fig a, right). The above two types of analysis showed a similar developmental path of epithelial cells in Monocle analysis. Furthermore, we showed the DDRtree plot for each patient and found that none of the patients dominated in specific state, which means that there is no large batch effect in this analysis. (please refer to the following Fig b, which is shown in the new version of Fig s7d).

a. Monocle analysis showed that distribution of epithelial cells in different setting of parameters.

b. Monocle analysis showed the distribution of epithelial cells on DDRtree in each patient

3) what is the actual number of cell per cluster, per patient and per tissue?

Response:

As suggested, we have provided precise numbers of cells per cluster, per patient and per tissue in supplementary table 1.

4) what are the R values in fig 3 d and e?

Response:

In the previous version, we calculated average score of representative genes for each cluster in each patient and each dot represents average score of patients of corresponding cluster. In the new figure, each dot represents corresponding cluster for each patient. Pearson correlation and linear regression were used. And R values have been updated in Fig 3d & e. Corresponding description has been updated in figure legend of Fig 3d & e.

5) what are the units of the axis in figure 4c-e: UMIs?

Response:

The units of the axis in figure 4c-e are normalized expression of representative genes, which have been described in the new version of figure legend.

6) how are “exhausted”, “resident memory” and the other diffusion axes defined?

Response:

The diffusion axes were annotation with canonical CD8 T signature genes including exhausted (TIGIT), resident memory (CD69) and naïve (SELL) (Fig 2b), which have been described in page 6 paragraph 2.

7) The relevance of results shown in figure 3f is difficult to appreciate.

Response:

We agree with the reviewer that the results shown in fig. 3f have little significance and do not add substantially to the manuscript, therefore, we deleted this figure in the revised version.

8) The conclusion that tumor T cells are more activated than LN T cells because there are less naïve cells in the tumor, does not consider that naïve T cells are normally present in LNs (even in healthy individuals).

Response:

We agree with the reviewer that there is indeed a large number of naïve T cells in the lymph node of LNMT, but the presence of tumor cells will change the microenvironment of lymph nodes, leading to maturation of Naïve T into active T cells. However, we found that T cells in LNMT are still not developed. To further answer this question, we firstly need to prove that the T cells collected from LN we analyzed are intra-tumoral. We performed spatial transcriptomics in the LN tissues of 4 patient samples which have been analyzed by scRNA-Seq. To integrate the data, scRNA-seq transcriptome and spatial transcriptome have been preprocessed by SCTransform function and PCA analysis. Then FindTransferAnchors and TransferData function were used to measure each cluster score for each spot. Description of methods has been updated in page 21 paragraph 3.

The data of the spatial transcriptomics showed that the ratio of tumor cells is over 50% in all the 4 patients and over 80% in three of them, suggesting that the samples are from LN metastases (Fig. 2 below which have been provided in the new version of Fig. 1f).

Corresponding descriptions in the Results part have been added on page 5, paragraph 1.

Further we analyzed the distribution of immune cells in LNMT and found that a large amount of immune cells are infiltrated around tumor cells, which are mainly plasma B, CD4 T and CD8 T cells. However, we found that most of T cells in LNMT are Naïve or in early developmental state (Fig. c). We then mapped our single cell sequencing signature into spatial transcriptome and confirmed that these naïve T cells have low activity. In conclusion, the data of spatial transcriptomics proved that the samples thrown to single cell sequencing are mainly from lymph node metastasis and further demonstrated that T cells around tumors in LN are less active (please refer to the figure 2 below).

Fig a. Spatial transcriptome integrated with scRNA-seq revealing that distribution of epithelial cells in LNMT.

Fig b. Bar plots showed that percentage of spots in spatial analysis with tumor cells in LNMT.

Fig c Spatial transcriptome analysis showed the distribution of epithelia cell, naïve CD4 T cells (CD4-C1-RPL) and naïve CD8 T cells (CD8-C1-CD8D) in LNMT.

9) The text should be reviewed by a native English speaker, as it contains many mistakes.

Response:

We are sorry for the mistakes. As suggested, the manuscript has been read and edited by a native English speaker, we hope it has been improved.

REVIEWER COMMENTS

Reviewer #2 (Remarks to the Author):

The authors have made substantial efforts which have improved the manuscript.

Their responses have addressed several of my major concerns (numbered according to the rebuttal):
2,3,4,5

Regarding Point 8, I am happy with the rewording. However, the authors should use paired analysis to look at gene expression differences between matched T cells in PT vs LN. It appears that they have not done this. I.e, compare the GE differences between each matched cell pair, then combine across all the pairs for each cell type.

However, I have two major concerns that are not addressed by these efforts and that I feel are fundamental to this manuscript.

1) I am still concerned that the LN dataset is heavily contaminated by immune cells from remnants of the lymph node adjacent to the tumor. I don't dispute that there are metastatic deposits in these LNs, but rather that there are also regions of non-invaded LNs. These regions are densely packed with immune cells and so will be highly represented in the scRNA data. The data presented can't distinguish whether the immune cell populations discussed are genuinely in the metastatic lesion, or distant from it in the remains of the LN.

The new Spatial Transcriptomics data is a nice addition to the dataset, but it appears to support my concern. For instance, the spatial heatmap shown (on Page 3 of the rebuttal) shows the naïve T cell signature largely enriched in regions distant from epithelial cells.

For this reason, many of the key results in the manuscript regarding differences in immune cells between PT and LN, both T cells & myeloid, are likely impacted by this caveat. These data underpin a major proportion of this paper's major findings.

2) The application of pseudotime analysis to epithelial cells is still very unsatisfying. The issue is not batch correction, rather the entire rationale. Batch correction of cancer cells will introduce major artefacts, because the cancer cells between patients are actually very different and should not be combined.

Pseudotime analysis was developed for developmental systems, where cells are genuinely transitioning in real time. That is not the case here- the cancer cells have probably been evolving for 10 years from their normal counterpart. So how do we interpret these results?

Finally, even if we accept the validity of this method, the findings appear to be entirely facile: namely that normal cells form one pseudotime population and cancer cells comprise the other 2, which are primarily based on disease subtype. This does not provide any biological insight.

Reviewer #3 (Remarks to the Author):

I would like to thank the authors and I am satisfied with the revisions. I hope this will be a great resource for all cancer researchers.

Reviewer #4 (Remarks to the Author):

The authors have made a big effort to address the multiple remarks and performed extra experiments, notably the spatial transcriptomics analysis, all what have highly improved the overall quality of the manuscript.

Reviewer #4 (additional comments):

Reviewer's #4 comments:

Indeed, the conclusion that “LNMT T cells are less mature than tumor T cells” is intrinsically affected by the naïve T cells that are normally present in LNs (even in healthy individuals), and by the fact that naïve T cells do not actively migrate to the tumor. The new data on the spatial transcriptome of the LNs is nice, and confirm reviewer #2 and my thoughts, that there are still considerable amounts of naïve T cells in the LNs (what is intuitively expected). The authors conclude that the spatial data “further demonstrated that T cells around tumors are less active”, but do not comment on the large fraction of naïve cells detected. To my understanding, the authors processed a piece of LNMT, and did not do laser microdissection to take the “tumor invaded zone” of the LNs, then, their statement: “further prove that the cells we collected were intratumoral “is an overstatement.

A different way of comparing the state of the cells between the LNMT and the PT would be to take cells from the same clone (same TCR) present in the LNMT vs in the PT and compare their transcriptome. Still, they should have enough cells to do the analysis.

The conclusion that “CD8+ T cells in PTs have a higher activation score than those in LNMTs” is not novel, as it has been already quite documented without needing to recur to single cell sequencing.

Their description on “5 CD4+CXCL13+ T cells” (maybe they should write 5 clusters of CD4) is quite superficial and with no functional validation, they do rapid extrapolations with some references to the literature, which are interesting as hypotheses but suffer from no validation (FACS, identification in the spatial space...). Mainly I refer to this paragraph:

“Cluster 2 expressed a higher level of interferon gamma (IFN- γ) than did the other CD4+CXCL13+ clusters, indicating that this type of CD4 cell is primarily responsible for killing tumor cells (Fig. 2i). Intriguingly, we found that BHLHE40 was highly expressed in cluster 2 (Fig 2i), which is consistent with a previous study that reported BHLHE40+CD4+ T cells to have the ability to suppress colon cancer cells, further supporting the tumor-suppressing function of cluster 2.’

The methodology is sometimes difficult to evaluate, and I am afraid that some premises are initially ill defined, as is the case of the presence of naïve T cells in LNMT or the unclear boundaries between cell identity and cell states in the migration chapter. Trajectory analysis could earn solidity by treating each patient separately and showing that conclusions obtained with all patients pooled are validated. Data are sometimes treated superficially, not biologically validated, and the conclusion are quite expected.

Nevertheless, the authors have made a considerable work, the overall described data is very valuable and generates interesting hypothesis. In any case, the pitfalls of the study should be well identified and discussed in the manuscript, if it is accepted for publication.

Reviewer #2 (Remarks to the Author):

The authors have made substantial efforts which have improved the manuscript.

Their responses have addressed several of my major concerns (numbered according to the rebuttal): 2,3,4,5

1. Regarding Point 8, I am happy with the rewording. However, the authors should use paired analysis to look at gene expression differences between matched T cells in PT vs LN. It appears that they have not done this. I.e., compare the GE differences between each matched cell pair, then combine across all the pairs for each cell type.

Response:

Thanks for the suggestion. We have used paired analysis to determine gene expression differences between matched T cells in PT vs LN. Please refer to the following detailed results and response in the question 2.

However, I have two major concerns that are not addressed by these efforts and that I feel are fundamental to this manuscript.

2. I am still concerned that the LN dataset is heavily contaminated by immune cells from remnants of the lymph node adjacent to the tumor. I don't dispute that there are metastatic deposits in these LNs, but rather that there are also regions of non-invaded LNs. These regions are densely packed with immune cells and so will be highly represented in the scRNA data. The data presented can't distinguish whether the immune cell populations discussed are genuinely in the metastatic lesion, or distant from it in the remains of the LN.

The new Spatial Transcriptomics data is a nice addition to the dataset, but it appears to support my concern. For instance, the spatial heatmap shown (on Page 3 of the rebuttal) shows the naïve T cell signature largely enriched in regions distant from epithelial cells.

For this reason, many of the key results in the manuscript regarding differences in immune cells between PT and LN, both T cells & myeloid, are likely impacted by this caveat. These data underpin a major proportion of this paper's major findings.

Response:

Thanks for this important comment and very good question. Indeed, it is very difficult to absolutely exclude the potential contamination with adjacent normal LN tissue, and it is a common problem in the existing studies of scRNA-Seq. At present, to strengthen our conclusion that immune cell activity of metastatic LNs is repressed compared with PT, and to maximumly reduce the interference by the potential contamination, we have taken three ways as shown below:

1) In the previous version, by applying matched T analysis, we have compared several marker genes for matched T cell within each T cell clusters. Because shared T cells may have different states in our dataset, comparison of shared T cells within each cluster may ignore the difference in T cell

state. To explore this deeply, we divided T cells into shared CD8 T cells and shared CD4 T cells, which were then calculated the transcriptome changes between PT and LNMT. We found that shared CD8 T cells in PT show higher expression of IFNG and CCL4L2, which correspond to higher tumor-killing function in PT. Whereas, shared CD8 T cells in LNMT show higher expression of RPL/RPS genes, which correspond to naïve signature (Fig. 1a). For CD4 T cells, we found that shared CD4 T cells in LNMT have higher expression of FOXP3, a marker gene of Treg, suggesting stronger immune repression in LNMT (Fig. 1b). These results indicated again that T cells are more mature in PT than in LNMT.

2) As suggested, we also compared the gene expression differences between matched T cells in PT vs LN using paired analysis to confirm our conclusion. We found that FOXP3 of matched CD4 T cell pairs was upregulated in LNMT compared with PT. CXCL13 and NMB of matched CD4 T cell pairs were upregulated in PT than in LNMT (Fig. 1c). For CD8 T cell pairs, IFNG was upregulated in PT cells and TGFB1 was increased in LNMT (Fig. 1d). Due to the sparsity and low sequencing depth of single cell technology, some genes expression differences were not significant in the comparisons. However, these analyses support our conclusion that T cells in PT showed higher tumor-killing ability and T cells in LNMT showed immune repression ability. Fig. 1c and Fig. 1d have been added into supplemental Fig 4c and 4d in the revised version. Corresponding description has been supplemented in page 9 paragraph 2.

Fig.1a Comparison of shared CD8+ T cells between LNMT and PT.

Volcano plot showing the differentially expressed genes between PT and LNMT in matched CD8+ T cells. P value <.05, log₂(fold change) ≥ 0.5.

Fig.1b Comparison of shared CD4+ T cells between LNMT and PT. Volcano plot showing the differentially expressed genes between PT and LNMT in matched CD4+ T cells. P value <.05, log₂(fold change) ≥ 0.5.

Fig.1c Comparison of shared CD4+ T cell clonal types between LNMT and PT by paired analysis.

Fig. 1d Comparison of shared CD8+ T cell clonal types between LNMT and PT by paired analysis.

3) We re-analyzed the reported scRNA-Seq data of 5 breast cancer patients (Kun Xu, et al. *Oncogenesis*, 2021), which include three groups of normal LN tissues, metastatic LN tissues and primary tumors (GSE180286). The cells with high expression of CD45 were selected as immune cells and their cell types were then defined by singleR methods (Fig 2a). Metastatic LN tissue displayed an inter-state between normal LN tissue and tumor in the components of cell types (Fig 2b). Main components of LN tissue were B cells and T cells. We collected T cells and separated them with a very high resolution to detail their similarities as much as possible. T cells were divided into 21 clusters and most of their distributions could be found in the three different groups (Fig 2c, 2d, 2e). To prevent the interference that the reviewer mentioned about non-invaded LN tissue, we removed the clusters only located in normal LN and metastatic LN whereas not existed in primary tumors, and compared difference of the remaining T cells between metastatic LN and primary tumor (Fig 2f). Consistent with our findings above, the remaining T cells in metastatic LN tissue have a higher RPL related gene expression compared with primary tumor, representing a more immature state in LN metastasis (Fig 2g, 2h)

We also analyzed all the clusters of myeloid cells using the data of GSE180286. The clusters of myeloid cells including DC cells and macrophages were shown in Fig 2i & 2j, then we compared them in metastatic LN and primary tumor. Consistent with our results, DC cells showed a higher expression of CXCL9 in PT compared with metastatic LN (Fig 2k), and macrophages in PT displayed a higher IFI27 expression (Fig 2l). Due to the relatively lower composition of myeloid cells in LN, we proposed that the interference of non-invaded LN in myeloid cells were much minor than in T cells.

Fig. 2 Analysis data of GSE180286

a. TSNE plot of immune cells from five primary tumors and ten paired axillary lymph nodes, colored according to cell types. b. Bar plot showing cellular composition of each cell type according to sample type. c. UMAP plot of T cells from five primary tumors and ten paired axillary lymph nodes,

colored according to cell types. d. UMAP plot of T cells from five primary tumors and ten paired axillary lymph nodes, colored according to cell clusters. e. Bar plot showing composition of sample type according to T cell clusters. f. UMAP plot of surplus T cells from five primary tumors and ten paired axillary lymph nodes, colored according to cell clusters. g. Volcano plot showing the differentially expressed genes between tumor and metastasis lymph node in surplus CD8+ T cells. P value < .05, log₂ (fold change) ≥ 0.5. h. Volcano plot showing the differentially expressed genes between tumor and metastasis lymph node in surplus CD4+ T cells. P value < .05, log₂ (fold change) ≥ 0.5. i. Bar plot showing composition of sample type according to DC cell clusters. j. Bar plot showing composition of sample type according to macrophage clusters. k. Volcano plot showing the differentially expressed genes between tumor and metastasis lymph node in DC cells. P value < .05, log₂ (fold change) ≥ 0.5. l. Volcano plot showing the differentially expressed genes between tumor and metastasis lymph node in macrophage. P value < .05, log₂ (fold change) ≥ 0.5.

3. The application of pseudotime analysis to epithelial cells is still very unsatisfying. The issue is not batch correction, rather the entire rationale. Batch correction of cancer cells will introduce major artefacts, because the cancer cells between patients are actually very different and should not be combined. Pseudotime analysis was developed for developmental systems, where cells are genuinely transitioning in real time. That is not the case here- the cancer cells have probably been evolving for 10 years from their normal counterpart. So how do we interpret these results?

Finally, even if we accept the validity of this method, the findings appear to be entirely facile: namely that normal cells form one pseudotime population and cancer cells comprise the other 2, which are primarily based on disease subtype. This does not provide any biological insight.

Response:

Thanks for the important advice. We agree with the reviewer that it is inappropriate to integrate cancer cells from different patients to apply pseudotime analysis. Also, it may not provide important biological insight when combining tumor cells together with normal cells into pseudotime analysis. As the reviewer mentioned, tumor cells may evolve for several years from their normal counterpart, which may be dramatically changed both in genomics and in transcriptions. Therefore, we removed the results in the last version about the application of pseudotime analysis to epithelial cells. We think it is interesting to investigate the features of tumor cells metastasizing from PT to LNMT for each patient. For his purpose, we re-analyzed the characteristics of malignant cells by CNV analysis and pseudotime analysis for each patient. The relationship between clonal types and malignant cell metastasis was investigated. We also analyzed the transcriptome changes of malignant cells after metastasizing to LNMT. The former figure 7 was replaced in the new version (please also refer to the figure 7 below). The description of Result is shown in page 13 and 14 marked in red.

Reviewer #3 (Remarks to the Author):

I would like to thank the authors and I am satisfied with the revisions. I hope this will be a great resource for all cancer researchers.

Reviewer #4 (Remarks to the Author):

The authors have made a big effort to address the multiple remarks and performed extra experiments, notably the spatial transcriptomics analysis, all what have highly improved the overall quality of the manuscript.

Reviewer #4 (additional comments):

Reviewer's #4 comments:

1. Indeed, the conclusion that “LNMT T cells are less mature than tumor T cells” is intrinsically affected by the naïve T cells that are normally present in LNs (even in healthy individuals), and by the fact that naïve T cells do not actively migrate to the tumor. The new data on the spatial transcriptome of the LNs is nice, and confirm reviewer #2 and my thoughts, that there are still considerable amounts of naïve T cells in the LNs (what is intuitively expected). The authors conclude that the spatial data “further demonstrated that T cells around tumors are less active”, but do not comment on the large fraction of naïve cells detected. To my understanding, the authors processed a piece of LNMT, and did not do laser microdissection to take the “tumor invaded zone” of the LNs, then, their statement: “further prove that the cells we collected were intratumoral “is an overstatement.

A different way of comparing the state of the cells between the LNMT and the PT would be to take cells from the same clone (same TCR) present in the LNMT vs in the PT and compare their transcriptome. Still, they should have enough cells to do the analysis.

Response:

Thanks for the important comment and suggestion. Indeed, it is very difficult to absolutely exclude the potential contamination with adjacent normal LN tissue. And it may be a common problem in the existing studies of scRNA-Seq. For the spatial transcriptome, just as the reviewer mentioned, we processed a piece of LNMT, and did not do laser microdissection to take the “tumor invaded zone” of the LNs. Therefore, we agree with the reviewer that the statement: “further prove that the cells we collected were intratumoral” is an overstatement. “further prove that the cells we collected were intratumoral” in page 5 was replaced by "further prove that most of the cells we collected were intratumoral".

At present, to strengthen our conclusion that immune cell activity of metastatic LNs is repressed compared with PT, and to maximumly reduce the interference by the potential contamination, we have taken three ways as shown below:

1) As suggested, the transcriptome of T cells from the same clone (same TCR) present in the LNMT vs in the PT was compared. We divided T cells into shared CD8 T cells and shared CD4 T cells, which were then calculated the transcriptome changes between PT and LNMT. We found that shared CD8 T cells in PT show higher expression of IFNG and CCL4L2, which correspond to higher tumor-killing function in PT. Whereas, shared CD8 T cells in LNMT show higher expression of RPL/RPS genes, which correspond to naïve signature (Fig. 1a). For CD4 T cells, we found that shared CD4 T cells in LNMT have higher expression of FOXP3, a marker gene of Treg, suggesting stronger

immune repression in LNMT (Fig. 1b). These results indicated again that T cells are more mature in PT than in LNMT.

2) We also compared the gene expression differences between matched T cells in PT vs LN using paired analysis to confirm our conclusion. We found that FOXP3 of matched CD4 T cell pairs was upregulated in LNMT compared with PT. CXCL13 and NMB of matched CD4 T cell pairs were upregulated in PT than in LNMT (Fig. 1c). For CD8 T cell pairs, IFNG was upregulated in PT cells and TGFB1 was increased in LNMT (Fig. 1d). Due to the sparsity and low sequencing depth of single cell technology, some genes expression differences were not significant in the comparisons. However, these analyses support our conclusion that T cells in PT showed higher tumor-killing ability and T cells in LNMT showed immune repression ability. Fig. 1c and Fig. 1d have been added into supplemental Fig 4c and 4d in the revised version. Corresponding description has been supplemented in page 9 paragraph 2.

Fig.1a Comparison of shared CD8+ T cells between LNMT and PT.

Volcano plot showing the differentially expressed genes between PT and LNMT in matched CD8+ T cells. P value <.05, log2(fold change) ≥ 0.5.

Fig.1b Comparison of shared CD4+ T cells between LNMT and PT. Volcano plot showing the differentially expressed genes between PT and LNMT in matched CD4+ T cells. P value $<.05$, $\log_2(\text{fold change}) \geq 0.5$.

Fig.1c Comparison of shared CD4+ T cell clonal types between LNMT and PT by paired analysis.

Fig. 1d Comparison of shared CD8+ T cell clonal types between LNMT and PT by paired analysis.

3) We re-analyzed the reported scRNA-Seq data of 5 breast cancer patients (Kun Xu, et al. *Oncogenesis*, 2021), which include three groups of normal LN tissues, metastatic LN tissues and primary tumors (GSE180286). The cells with high expression of CD45 were selected as immune cells and their cell types were then defined by singleR methods (Fig 2a). Metastatic LN tissue displayed an inter-state between normal LN tissue and tumor in the components of cell types (Fig 2b). Main components of LN tissue were B cells and T cells. We collected T cells and separated them with a very high resolution to detail their similarities as much as possible. T cells were divided into 21 clusters and most of their distributions could be found in the three different groups (Fig 2c, 2d, 2e). To prevent the interference that the reviewer mentioned about non-invaded LN tissue, we removed the clusters only located in normal LN and metastatic LN whereas not existed in primary tumors, and compared difference of the remaining T cells between metastatic LN and primary tumor (Fig 2f). Consistent with our findings above, the remaining T cells in metastatic LN tissue have a higher RPL related gene expression compared with primary tumor, representing a more immature state in LN metastasis (Fig 2g, 2h)

We also analyzed all the clusters of myeloid cells using the data of GSE180286. The clusters of myeloid cells including DC cells and macrophages were shown in Fig 2i & 2j, then we compared them in metastatic LN and primary tumor. Consistent with our results, DC cells showed a higher expression of CXCL9 in PT compared with metastatic LN (Fig 2k), and macrophages in PT displayed a higher IFI27 expression (Fig 2l). Due to the relatively lower composition of myeloid cells in LN, we proposed that the interference of non-invaded LN in myeloid cells were much minor than in T cells.

Fig. 2 Analysis data of GSE180286

a. TSNE plot of immune cells from five primary tumors and ten paired axillary lymph nodes, colored according to cell types. b. Bar plot showing cellular composition of each cell type according to sample type. c. UMAP plot of T cells from five primary tumors and ten paired axillary lymph nodes, colored according to cell types. d. UMAP plot of T cells from five primary tumors and ten paired axillary lymph nodes, colored according to cell clusters. e. Bar plot showing composition of sample type according to T cell clusters. f. UMAP plot of surplus T cells from five primary tumors and ten paired axillary lymph nodes, colored according to cell

clusters. g. Volcano plot showing the differentially expressed genes between tumor and metastasis lymph node in surplus CD8+ T cells. P value < .05, log₂ (fold change) ≥ 0.5. h. Volcano plot showing the differentially expressed genes between tumor and metastasis lymph node in surplus CD4+ T cells. P value < .05, log₂ (fold change) ≥ 0.5. i. Bar plot showing composition of sample type according to DC cell clusters. j. Bar plot showing composition of sample type according to macrophage clusters. k. Volcano plot showing the differentially expressed genes between tumor and metastasis lymph node in DC cells. P value < .05, log₂ (fold change) ≥ 0.5. l. Volcano plot showing the differentially expressed genes between tumor and metastasis lymph node in macrophage. P value < .05, log₂ (fold change) ≥ 0.5.

2. The conclusion that “CD8+ T cells in PTs have a higher activation score than those in LNMTs” is not novel, as it has been already quite documented without needing to recur to single cell sequencing.

Response:

We agree with the reviewer that “CD8+ T cells in PTs have a higher activation score than those in LNMTs” is not novel, we have revised the description in page 7 paragraph 1. However, we think the novelty of our study in analyzing CD8+ T cells between PT and LNMT is that we identified the phenotype and characteristics of CD8+ T cells within the two tissues in detail, such as the subpopulation and development path of CD8 T cells in PT vs LNMT, which can be well characterized in single cell sequencing.

3. Their description on “5 CD4+CXCL13+ T cells” (maybe they should write 5 clusters of CD4) is quite superficial and with no functional validation, they do rapid extrapolations with some references to the literature, which are interesting as hypotheses but suffer from no validation (FACS, identification in the spatial space...). Mainly I refer to this paragraph:

“Cluster 2 expressed a higher level of interferon gamma (IFN- γ) than did the other CD4+CXCL13+ clusters, indicating that this type of CD4 cell is primarily responsible for killing tumor cells (Fig. 2i). Intriguingly, we found that BHLHE40 was highly expressed in cluster 2 (Fig 2i), which is consistent with a previous study that reported BHLHE40+CD4+ T cells to have the ability to suppress colon cancer cells, further supporting the tumor-suppressing function of cluster 2.’

Response:

Thanks for the comments. As suggested, “5 CD4+CXCL13+ T cells” has been revised into “5 clusters of CD4+CXCL13+ T cells”. Indeed, the description on “5 clusters of CD4+CXCL13+ T cells” is lack of validation and not solid. We have modified the description in page 7 paragraph 2 to make them more appropriate. We also emphasized it in the Discussion.

4. The methodology is sometimes difficult to evaluate, and I am afraid that some premises are initially ill defined, as is the case of the presence of naïve T cells in LNMT or the unclear boundaries between cell identity and cell states in the migration chapter. Trajectory analysis could earn solidity by treating each patient separately and showing that conclusions obtained with all patients pooled are validated. Data are sometimes treated superficially, not biologically validated, and the conclusion are quite expected.

Response:

Thank this reviewer for the advice. For figure 7, as suggested, we re-analyzed the characteristics of malignant cells by CNV analysis and pseudotime analysis for each patient. The relationship between clonal types and malignant cell metastasis was investigated. We also analyzed the transcriptome changes of malignant cells after metastasizing to LNMT. The former figure 7 was replaced in the new version (please also refer to the figure 7 below). The description of Result is shown in pages 13 and 14 marked in red.

5. Nevertheless, the authors have made a considerable work, the overall described data is very valuable and generates interesting hypothesis. In any case, the pitfalls of the study should be well identified and discussed in the manuscript, if it is accepted for publication.

Response:

Thank this reviewer again for the great encouragement and constructive advice to help us improving our manuscript.

Reviewers' comments:

Reviewer #2 (Remarks to the Author):

I appreciate that the authors have made genuine attempts to address my concerns and have added new data. These revisions go part way to addressing my main concern of contamination of the dataset with non-tumour associated immune cells from non-involved regions of the lymph node

The paired analysis shown in Rebuttal figure 1c&d does confirm some of the results observed using non-paired analysis, though it appears that only a handful of genes are now significant

The authors also analyse a second dataset comprised of normal and metastatic lymph node, in an effort to filter out non-cancer related cells from their dataset. This is a good strategy. However, it appears that few of the genes that were significant in the original analysis remain significant. The authors claim that the differential expression of "RPL related genes" confirms a more immature state of T cells in the metastases. I am concerned that this conclusion is now based on so few genes, and am also not familiar with the literature showing that RPL-related genes are well-accepted markers of T cell maturity or activation. As such I am not certain how strongly this new analysis supports the central proposition that T cells are suppressed in mLN.

The parallel analysis of myeloid cells reveals that un-involved lymph nodes contain very few myeloid cells relative to a mLN. As such, I am comfortable to accept that the myeloid analysis originally included in the paper is unlikely to be substantially compromised by contamination from normal lymph node.

Finally, I find that the 'new' final figure could be improved.

I don't believe that panels d&e are valuable, as they simply show modest changes in copy number changes in certain clones in specific patients. This is expected and has been reported previously.

I also find the use of pseudotime in panel F to still be inappropriate and to add little to the paper, as I don't think pseudotime should be used on aggregated cells across different lesions.

However, The DE analysis comparing mLN to PT is meaningful and I suggest be incorporated into the main figure as panel C

Furthermore, I recommend also performing DE analysis to determine whether CNV clones express different genes when in the PT vs mLN

Reviewer #4 (Remarks to the Author):

Third round of revision,

The authors have performed extra analysis, which nevertheless still have some problems.

First, concerning the comparison of T cells between the LNMT and PT, the authors have used different approaches.

In the first approach, they have compared shared CD8 T cells and shared CD4 T cells.

A major concern is that the authors “re-bulk” the single cells (what is necessary to do DEG when using scRNAseq data), but they do a unique CD4+ T cell group. Thus, they are putting together Tregs and Tconvs as one cluster entity, which is conceptually incorrect. Consequently, their conclusions are likely biased by the percentage of shared CD4+ Tregs and CD4+ Tconvs in each organ. Indeed, FOPXP3 is one of the differentially expressed gene. Their conclusion:” suggesting stronger immune repression in LNMT (Fig. 1b). These results indicated again that T cells are more mature in PT than in LNMT” are likely a consequence of the chosen strategy and may not reflect the underlying biology.

Also, the total numbers of up or downregulated genes are not shown and the authors showcase some genes: are shown genes all the DEGs? Or only a selection? If they are numerous, doing pathways analysis should be less biased than showing some selected genes (I do not mean that this should be done here, as I consider that the initial clustering is not correct).

In the second approach, they use paired analysis between matched T cells. Here, the problem is that scRNAseq has many dropout genes, so DEGs between individual cells are highly inconsistent (as can be observed from their figures). Analysis may require a further step of re-assigning the T cells with matched TCRs to the initially defined clusters, re-bulking them by cluster, and then doing the DEGs (as an integrative strategy of the 3 studies proposed here by the authors). Moreover, the shown analysis is biased by the selection of a few genes and the conclusion that genes (FOXP3, IFNG, TGFBI) are up or down regulated, when they show p values higher than 0.01.

I am afraid there may not be one straight forward way of analyzing the results, but definitely, clustering CD4+ T conv and Treg cells together does not seem a good approach.

Overall, the data indicates that LNMT and PT have different proportions of T cell subpopulations in different states. Cells with shared TCRs may be found in the LNMT and PT, as part of the process of initial priming in the LNs and then migrating to the tumor, where they get re-activated upon re-encounter with their cognate antigen. Then, we should be careful with the interpretation, that a T cell

expresses more LAG3, TIGIT, TOX2; etc, may not indicate “more functional”, but more “exhausted”. And concluding that T cells are more mature in the PT than in the LNMT is somehow expected. More FOXP3 expression in the LNs, may just reflect more Tregs, not necessarily an “stronger immune repression” (maybe these Tregs in the LNs are not reactive to tumor specificities”,

Concerning my last remark:

4. The methodology is sometimes difficult to evaluate, and I am afraid that some premises are initially ill defined, as is the case of the presence of naïve T cells in LNMT or the unclear boundaries between cell identity and cell states in the migration chapter. Trajectory analysis could earn solidity by treating each patient separately and showing that conclusions obtained with all patients pooled are validated.

I am referring to the T cell part, not the tumor part, that was the question of the previous reviewer.

Reviewers' comments:

Reviewer #2 (Remarks to the Author):

I appreciate that the authors have made genuine attempts to address my concerns and have added new data. These revisions go part way to addressing my main concern of contamination of the dataset with non-tumour associated immune cells from non-involved regions of the lymph node

The paired analysis shown in Rebuttal figure 1c&d does confirm some of the results observed using non-paired analysis, though it appears that only a handful of genes are now significant

Response:

Thanks a lot for the reviewer's appreciation of our efforts. According to the reviewer's suggestion in the last revision, we used paired analysis to exclude possible contamination of resident naïve T in lymph nodes in LNMT vs PT analysis. Just as this reviewer mentions, it seems that the significant genes are not too many. We think the main reason is the limitation of paired analysis used for comparing the matched clonotypes in different tissue. scRNA-seq has many dropout genes, which means gene counts could not be captured completely, resulting in high variations of gene expression, especially for low abundance genes. In the paired analysis, we averaged the expression of genes for clonotypes to calculate DEGs. There are about 2-10 cells within each clonotype, averaging the expression of genes within each clonotype will artificially largely reduce the statistic numbers in the paired analysis, which may cause a more insignificant p-value compared with DEG analysis at the single-cell level. Although the method of paired analysis has its limitations using here, it has achieved the expected goal by partly confirming the results observed using non-paired analysis.

To comprehensively compare the differential genes in PT and LNMT, we included all matched CD8 T cells in DEG analysis. We found that 128 genes were significantly down-regulated and 63 genes were up-regulated in matched CD8 T cells of LNMT compared with PT (Figure 1a). These significant genes were then thrown into pathway enrichment analysis to uncover the underlying biological functions. We found that the down-regulated genes of matched CD8 T cells in LNMT are enriched in T cell activation and cytokine-mediated signaling pathway (Figure 1 b & c). Similar to CD8 T cells, matched conventional CD4 T (Tconvs) are less enriched in the T cell activation pathway and cytokine production pathway in LNMT than in PT (Figure 2). In summary, these data supported that T cells are less activated in LNMT compared with PT.

The former figures 4c-f and supplemental figures 4a-d were removed from the manuscript. Figure 1a,1c, 2a, and 2c below were added to figures 4c-f in the revised version. Figure 1b and 2b below were added to supplemental figure 4a&b in the revised version. Corresponding descriptions have been supplemented on page 9 paragraph 2.

Figure 1. **a.** Differential expression gene analysis of matched CD8 T cells for PT vs LNMT. **b.** GO term enrichment analysis for up-regulated genes of matched CD8 T cells in LNMT compared with PT. **c.** GO term enrichment analysis for down-regulated genes of matched CD8 T cells in LNMT compared with PT.

Figure 2. **a.** Differential expression gene analysis of matched Tconvs for PT vs LNMT. **b&c.** GO term enrichment analysis for up-regulated (b) and down-regulated (c) genes of matched Tconvs in LNMT compared with PT.

1) The authors also analyse a second dataset comprised of normal and metastatic lymph node, in an effort to filter out non-cancer related cells from their dataset. This is a good strategy. However, it appears that few of the genes that were significant in the original analysis remain significant.

Response:

Thanks for the reviewer’s reminder. We found that we set a very restricted threshold of percent expressed genes as 0.5 to do DEG analysis when analyzing the second dataset of 5 breast cancer patients (Kun Xu, et al. Oncogenesis, 2021, GSE180286) in the last revision, which means that genes expressed in less than 50% cells have been excluded. This results in a few significant genes containing 23 up-regulated and 17 down-regulated genes in LNMT. We adjusted the parameter to a routine one (0.1) and found that 31 genes were down-regulated and 61 genes were up-regulated in LNMT for CD8 T cells as shown in the below figure 3.

The significant genes in LNMT from this dataset were then used for pathway enrichment analysis and found that down-regulated genes in LNMT are also enriched in the pathway of T cell activation, which further supports our conclusion that T cells in mLN are less active than that in PT (Figure 4 below).

Figure 3. Differential expression genes analysis of matched CD8 T cells for PT vs mLN from GSE180286.

Figure 4. Pathway enrichment analysis for down-regulated genes in mLN vs PT in GSE180286.

2) The authors claim that the differential expression of “RPL related genes” confirms a more immature state of T cells in the metastases. I am concerned that this conclusion is now based on so few genes, and am also not familiar with the literature showing that RPL-related genes are well-accepted markers of T cell maturity or activation. As such I am not certain how strongly this new analysis supports the central proposition that T cells are suppressed in mLN.

Response:

Thanks for the critical question. “RPL-related genes” means ribosome-related genes. Several studies have revealed that ribosome genes are related to naïve T cells. A previous report demonstrated that naïve T cells have more abundant ribosome genes than other functional T cells (Ricciardi S, et al. Cell 2018). And large amounts of ribosome proteins maintain a prepared state for rapid immune response in naïve T cells (Tobias Wolf, et al. Nature Immunology 2020). In this study, we have found that there are 30 signature genes enriched in naïve T cells including CD8-C1-CD8B and CD4-C1-RPL defined by SELL and CCR7 (well-known markers of naïve T cells). Among the 30 signature genes, ribosome genes occupy a very high percentage with 11 of 30 for CD8-C1-CD8B and 25 of 30 for CD4-C1-RPL (Supplemental table 2 of manuscript or table 1 below). Furthermore, the online enrichment analysis tools (Toppgenes, <https://toppgene.cchmc.org/>) were applied to perform signature enrichment analysis to investigate the feature of up-regulated genes in LNMT vs PT. Among the significant genes, RPL genes contribute to naïve T signature mostly (Table 2). Importantly, we analyzed the correlation between RPL genes and naïve T cell markers (SELL&CCR7). The data showed that RPL genes are strongly positively correlated with SELL and CCR7 both in CD4 T cells and CD8 T cells (Figure 5 below). Taken together, our data revealed that RPL genes are a novel type of genes representing a more immature state of T cells.

We found that a series of ribosome genes are up-regulated in matched CD8 T cells of LNMT compared with PT by DEG analysis (figure 1a above), which supports the conclusion that T cells are more immature in mLN than PT.

Table 1. RPL-related genes are highly enriched in naïve T cells

avg_log2FC	p_val_adj	cluster	gene
0.71998072	8.34E-295	CD4-C1-RPL	RPL32
0.77002768	6.65E-270	CD4-C1-RPL	RPS13
0.67401072	1.06E-263	CD4-C1-RPL	RPL11
0.69993559	7.84E-262	CD4-C1-RPL	RPS8
0.62483116	1.16E-242	CD4-C1-RPL	RPS12
0.61844751	9.89E-236	CD4-C1-RPL	RPS14
0.60210254	5.68E-232	CD4-C1-RPL	RPL13
0.62264242	5.84E-215	CD4-C1-RPL	RPS23
0.66440235	7.59E-215	CD4-C1-RPL	RPS6
0.57918565	1.14E-208	CD4-C1-RPL	RPL19
0.52558866	3.74E-204	CD4-C1-RPL	RPL10
0.7278483	1.33E-203	CD4-C1-RPL	RPS5
0.53288947	1.71E-196	CD4-C1-RPL	RPL30
0.60011498	1.10E-195	CD4-C1-RPL	RPS4X
0.55708113	6.67E-191	CD4-C1-RPL	RPS27A
0.70710954	4.31E-188	CD4-C1-RPL	RPL22
0.65477884	1.76E-186	CD4-C1-RPL	RPS3A
0.59239622	3.90E-178	CD4-C1-RPL	RPL29
0.57707763	6.96E-178	CD4-C1-RPL	RPL18
0.5737648	4.06E-173	CD4-C1-RPL	RPS18
0.67883408	7.87E-164	CD4-C1-RPL	RPL5
0.52262802	7.51E-163	CD4-C1-RPL	RPL34
0.55084952	5.09E-157	CD4-C1-RPL	RPL3
0.50562984	1.73E-145	CD4-C1-RPL	RPL35A
0.50936226	1.20E-137	CD4-C1-RPL	RPL14
0.5481091	1.05E-194	CD8-C1-CD8B	RPS12
0.54758553	3.44E-186	CD8-C1-CD8B	RPL32
0.65608841	2.55E-181	CD8-C1-CD8B	RPS5
0.60391206	7.36E-173	CD8-C1-CD8B	RPS13
0.50727217	1.10E-172	CD8-C1-CD8B	RPS14
0.53343197	5.17E-167	CD8-C1-CD8B	RPS8
0.5123698	6.95E-163	CD8-C1-CD8B	RPS23
0.55696889	2.68E-146	CD8-C1-CD8B	RPS6
0.55293292	1.53E-121	CD8-C1-CD8B	RPL5
0.50189845	1.96E-107	CD8-C1-CD8B	RPL22

Table 2. The signature enrichment results of up-regulated genes in matched CD8 T cells of LNMT

Name of signature	Q value	Genes contribute to enrichment (Overlay of signature genes and input genes)
remission-CD8+ T naive remission / disease stage, cell group and cell class	6.999E-66	RPL21, RPL22, SELL, RPL32, RPL34, RPL37, RPL36A, RPS2, RPS3, RPS3A, NOP53, RPS4X, RPS6, ICAM2, RPS8, RPS10, RPS13, RPSA, RPS23, LDHB, PABPC1, PLAC8, AREG, EEF1A1, RPL10A, LIMD2, PASK, EEF1B2, JUNB, LTB, CCR7, BEX2, RPL3, RPL4, RPL5, RPL7, RPL9, RPL17, FCMR
C-Lymphocytic-CD4 T-cell-CD4+ Naive T cell / CD4+ Trm cell CD4 T-cell / Donor, Lineage, Cell class and subclass	1.094E-55	RPL21, RPL22, SELL, RPL32, RPL34, RPL37, RPL36A, LINC00402, RPS2, RPS3, RPS3A, NOP53, RPS4X, RPS6, RPS8, RPS13, RPSA, RPS23, LDHB, HCST, EEF1A1, RPL10A, GSTM3, PASK, EEF1B2, EEF1G, CD27, LTB, CCR7, RPL3, RPL5, RPL7, RPL9, RPL17, RPL7, RPL9, RPL17, FCMR

Figure 5 The correlation of ribosome genes with naïve marker genes in CD4 T cells (left) and CD8 T cells (right). Average expression of ribosome genes or naïve marker genes (SELL and CCR7) were calculated in each cell cluster for each sample. Each dot represents a T cell cluster for each patient.

In fact, in addition to RPL genes, the new pieces of evidence above figures 1-4 also support the central proposition that T cells are suppressed in LNMT. Moreover, we have demonstrated that CXCL13+CD8+ T cells are evidently less enriched in LNMT than in PT (Figure 1e in the Manuscript). CXCL13 has been reported correlated with the anti-tumor immune microenvironment and high CXCL13 indicates the efficacy of PD-1 checkpoint blockade (Yang M, et al., J Immunother Cancer. 2021). Interestingly, a recent study recognized CXCL13+CD8+ T cells as the tumor neo-antigen reactive cell types (Chunhong Zheng, et al. Cell 2022). Thus, the

sharp reduction of CXCL13+CD8+ T cells in LNMT may be one of the mechanisms mediating the suppressed function of T cells in LNMT.

Reference:

Zheng C, Fass JN, Shih YP, Gunderson AJ, Sanjuan Silva N, Huang H, Bernard BM, Rajamanickam V, Slagel J, Bifulco CB, Piening B, Newell PHA, Hansen PD, Tran E. Transcriptomic profiles of neoantigen-reactive T cells in human gastrointestinal cancers. *Cancer Cell*. 2022 Apr 11;40(4):410-423.e7. doi: 10.1016/j.ccell.2022.03.005. PMID: 35413272.

Chaput N, Darrasse-Jèze G, Bergot AS, Cordier C, Ngo-Abdalla S, Klatzmann D, Azogui O. Regulatory T cells prevent CD8 T cell maturation by inhibiting CD4 Th cells at tumor sites. *J Immunol*. 2007 Oct 15;179(8):4969-78. doi: 10.4049/jimmunol.179.8.4969. PMID: 17911581.

Ricciardi S, Manfrini N, Alfieri R, Calamita P, Crosti MC, Gallo S, Müller R, Pagani M, Abrignani S, Biffo S. The Translational Machinery of Human CD4+ T Cells Is Poised for Activation and Controls the Switch from Quiescence to Metabolic Remodeling. *Cell Metab*. 2018 Dec 4;28(6):961. doi: 10.1016/j.cmet.2018.09.010.

Wolf T, Jin W, Zoppi G, Vogel IA, Akhmedov M, Bleck CKE, et al. Dynamics in protein translation sustaining T cell preparedness. *Nat Immunol*. 2020; 21:927–37.

Yang M, Lu J, Zhang G, Wang Y, He M, Xu Q, Xu C, Liu H. CXCL13 shapes immunoactive tumor microenvironment and enhances the efficacy of PD-1 checkpoint blockade in high-grade serous ovarian cancer. *J Immunother Cancer*. 2021 Jan;9(1): e001136. doi: 10.1136/jitc-2020-001136. PMID: 33452206; PMCID: PMC7813306.

The parallel analysis of myeloid cells reveals that un-involved lymph nodes contain very few myeloid cells relative to a mLN. As such, I am comfortable to accept that the myeloid analysis originally included in the paper is unlikely to be substantially compromised by contamination from normal lymph node.

Response:

Thanks for the positive comment.

Finally, I find that the ‘new’ final figure could be improved.

I don’t believe that panels d & e are valuable, as they simply show modest changes in copy number changes in certain clones in specific patients. This is expected and has been reported previously. I also find the use of pseudotime in panel_F to still be inappropriate and to add little to the paper, as I don’t think pseudotime should be used on aggregated cells across different lesions. However, The DE analysis comparing mLN to PT is meaningful and I suggest be incorporated into the main figure as panel C Furthermore, I recommend also performing DE analysis to determine whether CNV clones express different genes when in the PT vs mLN.

Response:

Thanks for the comments and good suggestions. We agree with the reviewer’s opinion, panels d-f of the former figure 7 have been deleted. As the reviewer suggested, the DEG analysis was performed to compare the tumor cells in PT vs LNMT, and pathway enrichment analyses were conducted to uncover the underlying biological functions (Supplemental Table 4&5). Interestingly, we have novel and important findings that the antigen-presentation pathways were

down-regulated in malignant cells of the metastatic lymph node. The former figure 7 and supplemental figure 7 in the manuscript have been substantially updated in the new version (please also refer to figure 6 and the supplemental figure below), and the descriptions of the Result are shown on pages 13 and 14 marked in red. The detailed results are as below.

We found that antigen presentation genes, such as CD74, HLA-DRA and B2M, are mostly down-regulated in LNMT compared with PT for patient 8 (Figure 6c below). Similarly, we also found that HLA-B and HLA-C are down-regulated in LNMT for patient 5 (Supplementary figure e below). To character whether these findings are prevalent in most patients, transcriptome signatures from 5 patients were used for GSEA enrichment analysis. Significant pathways shared by 5 patients were analyzed and the normalized enrichment score (NES) for each pathway was averaged. The pathways were then ranked according to the numbers of shared patients and the mean of NES. Of the top 10 enriched pathways, 4 pathways are related to antigen presentation and these pathways are enriched in 4 of 5 patients (figure 6 d& e below, Supplemental Table 6).

It is interesting to ask whether down-regulated antigen presentation pathways are related to CNV clones of malignant cells, just as the reviewer suggested. We then clustered malignant cells according to their CNV similarity for each patient and then compared the DEG of each CNV clone in PT vs LNMT (Figure 6f below). We found that malignant cells belonging to different CNV clusters in patient 8 may not differ in metastasis ability (Figure 6g below). And the malignant cells in different CNV clusters of patient 8 are also enriched in the antigen presentation pathway (Supplemental Figure f & g below). Antigen presentation genes can be mainly divided into MHC I class molecules and MHC II class molecules. We compared the two types of antigen presentation in PT vs LNMT across the different CNV clusters and found that both MHC I class molecules and MHC II class molecules are down-regulated in LNMT across different CNV clusters in patient 8 (Figure 6h below). MHC I genes but not MHC II genes are down-regulated in LNMT of Patient 5 (Supplemental Figure h-j below). These findings indicate that malignant cells migrated to LNMT may render lower antigen presentation genes, resulting in an immune evasive mechanism, which provides new insight into the characterization of malignant cell metastasis in breast cancer.

Figure 6. **a.** A heatmap showing large-scale CNVs of epithelial cells (rows) from paired tumor tissues in 8 LNMT patients. CNVs in red indicate amplifications, while those in blue indicate deletions. CNVs were identified by inferCNV. **b.** the t-SNE plot of malignant epithelial cells identified by inferCNV malignant score. **c.** Volcano plot showing the differentially expressed genes between PT and LNMT in malignant cells from patient 8. P value < 0.05 , \log_2 (fold change) ≥ 0.5 . **d.** Barplot showed the top 10 normalized enrichment scores (NES) of the shared significantly enriched pathway across patients. Genes rank calculated by PT vs LNMT in malignant cells for each patient were used for NES calculating by GSEA analysis. Color represents the number of patients who are significant in the same pathway. **e.** GSEA analysis shows that genes rank in PT vs LNMT

of represented patients are enriched in the antigen presentation pathway. The left represents a high gene rank in PT. **f.** Circos plot showing 6 CNV clusters of malignant epithelial cells according to CNVs similarity for patient 8. Cells were colored by tissue. **g.** The bar plot shows the frequency of 6 CNV clusters in patient 8. **h.** Violin plots showing the MHC I antigen presentation signature (Up) and MHC II antigen presentation signature (Down) of 6 CNV clusters from patient 8.

Supplemental Figure. **a.** histogram showing the distribution of malignant scores in epithelial cells

and reference cells. Orange indicates the malignant score distribution of malignant epithelial cells, blue indicates nonmalignant epithelial cells, and grey indicates reference cells. **b.** Cells were classified into malignant epithelial cells, nonmalignant cells, and nonepithelial cells. A boxplot showing number of features (left) and epithelial scores (right) of the 3 types. **c, d.** The upset plot shows the overlapping of the upregulated genes in PT (c) and the upregulated genes in LNMT (d). Differential analysis was calculated for each patient. P value < 0.05 , \log_2 (fold change) ≥ 0.5 . **e.** Volcano plot showing the differentially expressed genes between PT and LNMT in malignant cells from patient 5. P value < 0.05 , \log_2 (fold change) ≥ 0.5 . **f-g.** Volcano plot showing the differentially expressed genes between PT and LNMT in malignant cells for CNV cluster 1(f) or CNV cluster 2 (g) from patient 8. P value < 0.05 , \log_2 (fold change) ≥ 0.5 . **h.** Circos plot showing 6 CNV clusters of malignant epithelial cells according to CNVs similarity for patient 5. Cells were colored by tissue. **i.** The bar plot shows the frequency of 5 CNV clusters in patient 5. **j.** Violin plots showing the MHC I antigen presentation signature (Up) and MHC II antigen presentation signature (Down) of 5 CNV clusters from patient 5.

Reviewer #4 (Remarks to the Author):

Third round of revision,

The authors have performed extra analysis, which nevertheless still have some problems.

First, concerning the comparison of T cells between the LNMT and PT, the authors have used different approaches.

In the first approach, they have compared shared CD8 T cells and shared CD4 T cells.

A major concern is that the authors “re-bulk” the single cells (what is necessary to do DEG when using scRNAseq data), but they do a unique CD4+ T cell group. Thus, they are putting together Tregs and Tconvs as one cluster entity, which is conceptually incorrect. Consequently, their conclusions are likely biased by the percentage of shared CD4+ Tregs and CD4+ Tconvs in each organ. Indeed, FOXP3 is one of the differentially expressed gene. Their conclusion:” suggesting stronger immune repression in LNMT (Fig. 1b). These results indicated again that T cells are more mature in PT than in LNMT” are likely a consequence of the chosen strategy and may not reflect the underlying biology.

Also, the total numbers of up or downregulated genes are not shown and the authors showcase some genes: are shown genes all the DEGs? Or only a selection? If they are numerous, doing pathways analysis should be less biased than showing some selected genes (I do not mean that this should be done here, as I consider that the initial clustering is not correct).

Response:

1) Thanks for the critical suggestion. We agree with the reviewer that it is inappropriate to integrate all shared CD4 T single cells in the differential analysis. Therefore, as suggested, CD4 T cells were separated into Tconvs and Tregs for differential analysis. The results showed that 460 genes were downregulated and 21 genes were upregulated in matched Tconvs of LNMT compared with PT (Figure 1a). To better understand the biological functions of the significant genes, GO term analysis was conducted for DEGs pathway enrichment. The analysis data revealed that the downregulated genes in Tconvs of LNMT are enriched in the T cell activation pathway and positive regulation of cytokine production (Figure 1b&c). As for Tregs, only 27 genes were down-regulated and 16 genes were up-regulated in the matched Tregs of LNMT compared with PT (Figure 2 a-c). Although Tregs have fewer significant genes compared with Tconvs, our previous analysis has discovered that the cell frequency of Tregs is more enriched in LNMT than in PT (Manuscript Figure 1e).

2) Furthermore, all matched CD8 T cells were used for DEG analysis. The results showed that 128 genes were significantly down-regulated and 63 genes were up-regulated in matched CD8 T cells of LNMT compared with PT (Fig 3a). These significant genes were then thrown into pathway enrichment analysis to uncover the underlying biological functions. We found that the down-regulated genes of matched CD8 T cells in LNMT are enriched in T cell activation, which also reflects a lower T cell activity in LNMT than in PT (Fig 3b-c).

In summary, the new data further supported that T cells are less activated in LNMT compared with PT.

The former figure 4c-f and supplemental figure 4a-d were removed from the manuscript. Figure 3a, 3c, 1a, and 1c below were added to figures 4c-f in the revised version. Figure 3b, 1b, and 2a

below were added to supplemental figure 4a-c in the revised version. Corresponding descriptions have been supplemented on page 9 paragraph 2.

Figure 1. **a**. Differential expression gene analysis of matched Tconvs for PT vs LNMT. **b&c** Pathway enrichment analysis for up-regulated (b) and down-regulated (c) genes of matched Tconvs in LNMT compared with PT.

Figure 2. **a** Differential expression gene analysis of matched Tregs for PT vs LNMT. **b&c** Pathway enrichment analysis for up-regulated (b) and down-regulated (c) genes of matched Tregs in LNMT compared with PT.

Figure 3. **a.** Differential expression gene analysis of matched CD8 T cells for PT vs LNMT. **b.** Pathway enrichment analysis for up-regulated genes of matched CD8 T cells in LNMT compared with PT. **c.** Pathway enrichment analysis for down-regulated genes of matched CD8 T cells in LNMT compared with PT.

In the second approach, they use paired analysis between matched T cells. Here, the problem is that scRNAseq has many dropout genes, so DEGs between individual cells are highly inconsistent (as can be observed from their figures). Analysis may require a further step of re-assigning the T cells with matched TCRs to the initially defined clusters, re-bulking them by cluster, and then doing the DEGs (as an integrative strategy of the 3 studies proposed here by the authors). Moreover, the shown analysis is biased by the selection of a few genes and the conclusion that genes (FOXP3, IFNG, TGFB1) are up or down regulated, when they show p values higher than 0.01.

I am afraid there may not be one straight forward way of analyzing the results, but definitely, clustering CD4+ T conv and Treg cells together does not seem a good approach.

Response:

Thanks for the reviewer’s suggestion. We agree with the reviewer that scRNA-seq has many dropout genes, resulting in high variations of gene expression, especially for low abundance genes, leading to few significant genes. According to the reviewer’s suggestion, we re-bulked the gene expression of matched CD8 T cells for each sample. Then samples were calculated for differential analysis in PT vs LNMT. The results showed that there are only 125 significant genes

among the total 18280 detected genes (Figure 4). Although re-bulking matched T cells can largely increase the gene abundance, the statistic numbers in each group for DEGs analysis were artificially dramatically reduced (7 PT vs 7 LNMT, matched T cells did not be captured in PT tissue of one patient), which led to very few significant genes (Figure 4). Thus, we think that re-bulking matched T cells may not be a good strategy.

As matched T may experience the same development period and can be distinguished from naïve T cells, the analysis of matched T cells is a good way to compare the difference between T cells in different tissue. Thus, the DEG analysis by combining with pathway enrichment shown above in figures 1-3 has already supplied evidence to support our conclusion that T cells in LNMT are less activated than that in PT.

Figure 4. Volcano plot showing the differentially expressed genes between PT and LNMT in re-bulk CD8 T cells. Gene expressions were re-bulked for CD8T cells for each sample. Gene expressions were normalized by the total library size in each sample. P value < 0.05 , \log_2 (fold change) ≥ 0.5 .

Overall, the data indicates that LNMT and PT have different proportions of T cell subpopulations in different states. Cells with shared TCRs may be found in the LNMT and PT, as part of the process of initial priming in the LNs and then migrating to the tumor, where they get re-activated upon re-encounter with their cognate antigen. Then, we should be careful with the interpretation, that a T cell expresses more LAG3, TIGIT, TOX2; etc, may not indicate “more functional”, but more “exhausted”. And concluding that T cells are more mature in the PT than in the LNMT is somehow expected. More FOXP3 expression in the LNs, may just reflect more Tregs, not necessarily an “stronger immune repression” (maybe these Tregs in the LNs are not reactive to tumor specificities”.

Response:

1) Thanks for the reviewer’s suggestion. Indeed, we should be careful with the interpretation to avoid overstating or misunderstanding. However, we are confused by the comment “a T cell expresses more LAG3, TIGIT, TOX2; etc, may not indicate “more functional”, but more

“exhausted”. In the last revision, we compared the gene expression differences between matched T cells in PT vs LN using paired analysis to confirm our conclusion. Although we analyzed the genes including LAG3, TIGIT and TOX2 (as shown below), there is no significance between PT and LN. And we did not state that T cells expressing more LAG3, TIGIT, and TOX2 indicate “more functional” in the manuscript or in the response letter.

2) In the last revision, we integrated all matched CD4 T cells into DEG analysis, which was considered a conceptual mistake by the reviewer. Thus, as this reviewer suggested, we separated CD4 T cells into Tregs and Tconvs to perform DEGs and pathway enrichment analysis. The details have been listed in the response to question 1. The previous figures of paired analysis in CD4 T cells have been replaced by new data. Thus, the queries about “LAG3, TIGIT, TOX2” and “FOXP3” the reviewer concerned have no longer existed.

Concerning my last remark:

4. The methodology is sometimes difficult to evaluate, and I am afraid that some premises are initially ill defined, as is the case of the presence of naïve T cells in LNMT or the unclear boundaries between cell identity and cell states in the migration chapter. Trajectory analysis could earn solidity by treating each patient separately and showing that conclusions obtained with all patients pooled are validated.

I am referring to the T cell part, not the tumor part, that was the question of the previous reviewer.

Response:

1) Thanks for the reviewer’s suggestion. Indeed, it is difficult to evaluate a study or a conclusion by one single methodology which may bias the conclusion. However, we have combined different methods to support our findings by providing multiple pieces of evidence. To resolve the query the reviewer’s concerns about the presence of naïve T cells in LNMT, several methods including spatial transcriptome, matched T analysis, and analysis of public dataset have been applied

together to avoid the interference of naïve T cells in our immune cell analysis. Nevertheless, it is exceedingly difficult to absolutely exclude normal naïve T cells in LNMT, which is a common problem existing in all studies of scRNA-seq. But we believe that very few naïve T cells in LNMT do not affect the conclusion of this study, as effective methods have been applied to exclude the potential interference.

As for the comment “the unclear boundaries between cell identity and cell states in the migration chapter”, we interpret it as “matched T cells may undergo different stages of differentiation, therefore, it may be not accurate to classify the matched T cells into many subclusters”. Thus, we removed the former version of matched T analysis. In this version, all matched CD8 T cells were used for DEG analysis, and CD4 T cells were classified into Tregs and Tconvs to perform DEGs, which were then for pathway enrichment analysis (shown in the above figure 1-3). We hope the new data can address the concern of the reviewer to the most extent.

2) As the reviewer suggested, we have performed trajectory analysis for 6 patients separately. The other 2 patients were excluded as they have few immune cells in PT. We found that the CD8 T cells trajectory path in each patient is compatible with the major trajectory from combining all CD8 T cells. As shown in figure 5, CD8-C1-CD8B differentiates into CD8-C2-CCL5 in patient 1, patient 2, patient 7 and patient 8. CD8-C2-CCL5 are connected to CD8-C3-GZMK or CD8-C4-HSPA1A in patient 1, patient 3, patient 7, and patient 8. In the final state, CD8-C3-GZMK are more likely connected to CD8-C5-CXCL13 in patient 3, patient 5, patient 6 and patient 7. Because of the random choice of the biopsy, each sample may lose some certain cell types, resulting in a bias of CD8 T cell types in different patients. Compared with doing trajectory path for each patient, we think integrating all CD8 T cells for analysis may depict the whole picture of CD8 T cells trajectory, which is a common way adopted in most studies of single-cell sequencing.

Figure 5 Diffusion maps were calculated for 6 representative patients.

REVIEWERS' COMMENTS

Reviewer #2 (Remarks to the Author):

I am satisfied that the authors have sufficiently addressed my concerns and in doing so improved the manuscript. I have no further issues to raise.